# Bi-allelic VPS16 variants limit HOPS/CORVET levels and cause a mucopolysaccharidosis-like disease

Kalliopi Sofou[1,†], Kolja Meier[2,†], Leslie E Sanderson[3,†], Debora Kaminski[4,5,6,†], Laia Montoliu-Gaya[4,6], Emma Samuelsson[5], Maria Blomqvist[4,5], Lotta Agholme[5,7], Jutta Gärtner[2], Chris Mühlhausen[2], Niklas Darin[1], Tahsin Stefan Barakat[3] 🔾, Lars Schlotawa[2], Tjakko van Ham[3] 🔾, Jorge Asin Cayuela[4,5] & Fredrik H Sterky[4,5,6,*] 🔾

## Abstract

Lysosomal storage diseases, including mucopolysaccharidoses, result from genetic defects that impair lysosomal catabolism. Here, we describe two patients from two independent families presenting with progressive psychomotor regression, delayed myelination, brain atrophy, neutropenia, skeletal abnormalities, and mucopolysaccharidosis-like dysmorphic features. Both patients were homozygous for the same intronic variant in *VPS16*, a gene encoding a subunit of the HOPS and CORVET complexes. The variant impaired normal mRNA splicing and led to an ~85% reduction in VPS16 protein levels in patient-derived fibroblasts. Levels of other HOPS/CORVET subunits, including VPS33A, were similarly reduced, but restored upon re-expression of VPS16. Patient-derived fibroblasts showed defects in the uptake and endosomal trafficking of transferrin as well as accumulation of autophagosomes and lysosomal compartments. Re-expression of VPS16 rescued the cellular phenotypes. Zebrafish with disrupted *ups16* expression showed impaired development, reduced myelination, and a similar accumulation of lysosomes and autophagosomes in the brain, particularly in glia cells. This disorder resembles previously reported patients with mutations in *VPS33A*, thus expanding the family of mucopolysaccharidosis-like diseases that result from mutations in HOPS/CORVET subunits.

**Keywords** autophagy; endosome; lysosomal storage disease; MPS; myelination
**Subject Categories** Genetics, Gene Therapy & Genetic Disease; Organelles

## Introduction

Lysosomal storage diseases (LSDs) are caused by monogenic defects in lysosomal enzymes, transporters, and membrane proteins (Platt *et al*, 2019; Parenti *et al*, 2021). LSDs typically present in childhood and primarily affect the central nervous system, although other organ systems are often also affected. Impaired lysosomal catabolism leads to the gradual buildup of undegraded macromolecules and their disease-specific patterns can be used to subclassify LSDs. For example, the mucopolysaccharidoses (MPSs) result from deficiencies in enzymes that degrade glycosaminoglycans (GAGs). Characteristic symptoms of the MPSs include coarse facial features, skeletal deformities, short stature, hepatosplenomegaly, cardiac abnormalities, and neurological impairments (Stapleton *et al*, 2018).

Recently, new genetic defects with LSD-like biochemical and clinical phenotypes have been identified in which the defective protein is not located in the lysosome itself but instead is involved in its turnover (Ballabio & Bonifacino, 2020). For example, a missense mutation in *VPS33A* found in Turkish and Yakutian populations causes a recessive condition that resembles an MPS with severe systemic involvement (Dursun *et al*, 2017; Kondo *et al*, 2017; Pavlova *et al*, 2019). The term "MPS-Plus" (OMIM #617303) has been coined to describe this syndrome. Furthermore, homozygous mutations in *VPS11* have been found in children presenting with leukoencephalopathy, seizures and joint contractures (Edvardson *et al*, 2015; Hörtnagel *et al*, 2016; Zhang *et al*, 2016). Intracellular accumulation of abnormal structures in storage macrophages in the bone marrow suggested lysosomal dysfunction (Hörtnagel *et al*, 2016). Moreover, excretion profiles of glycosphingolipids in VPS11 patients were reminiscent of *PSAP* deficiency (Hörtnagel *et al*, 2016), another LSD in which activator proteins (saposins) involved in the degradation of glycosphingolipids are deficient (Motta *et al*, 2016).

1  Department of Paediatrics, Institute of Clinical Sciences, University of Gothenburg, Gothenburg, Sweden
2  Department of Pediatrics and Adolescent Medicine, University Medical Center Goettingen, Goettingen, Germany
3  Department of Clinical Genetics, Erasmus University Medical Center Rotterdam, Rotterdam, The Netherlands
4  Department of Laboratory Medicine, Institute of Biomedicine, University of Gothenburg, Gothenburg, Sweden
5  Department of Clinical Chemistry, Sahlgrenska University Hospital, Gothenburg, Sweden
6  Wallenberg Centre for Molecular and Translational Medicine, University of Gothenburg, Gothenburg, Sweden
7  Department of Psychiatry and Neurochemistry, Institute of Neuroscience and Physiology, University of Gothenburg, Gothenburg, Sweden
   *Corresponding author. Tel: +46-317865357; E-mail: Fredrik.sterky@gu.se
   †These authors contributed equally to this work

VPS33A and VPS11 are part of the multi-subunit tethering complex HOPS (homotypic fusion and vacuole protein sorting) (Seals et al, 2000; Wurmser et al, 2000), which mediates fusion of lysosomes with late endosomes (also known as multivesicular bodies), phagosomes and autophagosomes (Luzio et al, 2007). HOPS additionally includes protein subunits VPS16, VPS18, VPS39, and VPS41 (Wickner, 2010; van der Beek et al, 2019) to form an elongated hetero-hexamer (Bröcker et al, 2012). For lysosomal membrane fusion, the HOPS subunits VPS39 and VPS41 tether the opposing target membranes, while the subunit VPS33A serves as an SM (Sec1/Munc18-like) protein (Südhof & Rothman, 2009) to catalyze membrane fusion by SNAREs (Baker & Hughson, 2016). The related CORVET (core Class C vacuole/endosome tethering) complex (Peplowska et al, 2007), with subunits VPS3 and VPS8 instead of VPS39 and VPS41, mediates in a similar manner fusion between early endosomes. In both complexes, VPS33A is recruited to the complex by direct binding to VPS16 (Baker et al, 2013; Graham et al, 2013; Wartosch et al, 2015). No LSD-like phenotypes have so far been linked to variants in VPS16. Instead, heterozygous loss-of-function mutations (Steel et al, 2020) as well as a homozygous missense variant found in a consanguineous Chinese family (Cai et al, 2016) have been found to cause dystonia with early-adolescent onset.

Here, we describe the first cases of an MPS-like disease caused by an intronic variant in VPS16. We identified two unrelated individuals presenting with a clinical phenotype characterized by global developmental regression, facial dysmorphism, delayed myelination, skeletal abnormalities (dysostosis multiplex), and recurrent anemia and neutropenia. The MPS-like dysmorphism, dysostosis multiplex, and vacuolizing lymphocytes were suggestive of lysosomal dysfunction, but biochemical and genetic tests for known LSDs were negative. Sequencing identified a homozygous intronic mutation in VPS16 that generates a new splice-acceptor site. Functional studies in fibroblasts showed reduced levels of VPS16 protein and other HOPS/CORVET complex subunits, defective transferrin uptake, and accumulation of lysosomal compartments and autophagosomes; phenotypes that were rescued upon re-expression of VPS16. A zebrafish model with disrupted vps16 expression similarly showed accumulation of autophagosomes and acidic compartments in the brain and demonstrated impaired myelination mimicking the clinical phenotype. Taken together, our results indicate that a reduction in VPS16 levels—and by extension HOPS/CORVET complexes—can cause a novel genetic MPS-like disorder.

## Results

### Case descriptions

For a detailed, chronological description of the patients' clinical phenotypes and diagnostic procedures, please see extended patient descriptions in the supplementary material. Main findings, in comparison with other cases of VPS subunit defects, are summarized in Table 1.

Patient A is a boy born at gestational week 34 as the fifth child to healthy first-cousin parents of Iranian descent (Family I; Fig 2A). He developed normally until the age of 9 months, when psychomotor skills regressed following a viral gastroenteritis. He then failed to attain new psychomotor skills and progressively developed central hypotonia and lower limb spasticity. He showed coarse facial features, which over time became more prominent with macroglossia and a thoracic deformity (Figs 1A and EV1). Brain MRIs showed a thin corpus callosum, abnormal myelination, and patchy white matter lesions in the periventricular and deep white matter (Fig 1D). MR spectroscopy of the white matter revealed a decreased N-acetylaspartate (NAA) peak (Fig 1F). The cerebellum had reduced volume, suggesting atrophy (Fig 1E). Contrast enhancements were seen at nerve roots of the conus medullaris (Fig 1G). EEG showed signs of generalized encephalopathy and neurophysiological testing confirmed a demyelinating sensorimotor polyneuropathy. Fundoscopic examination revealed mild pallor of the optic disks, suggesting atrophy. Biomarkers for brain damage in cerebrospinal fluid (CSF) were extremely elevated (tau protein 27,400 ng/l, ref < 250 ng/l; also see suppl material). Neutrophil counts were at or below the lower normal range ($0.3–4.3 \times 10^9$/l, ref $1.5–8.5 \times 10^9$/l). Vacuolized lymphocytes in blood were increased (8%, ref < 4%) and a bone marrow smear revealed myelopoietic cells with densely stained granules (Fig 1H), consistent with an LSD such as Chediak–Higashi syndrome. At an age of 3.5 years, psychomotor development had further regressed with severe lower limb spasticity and feeding problems necessitating a gastrostomy. Heart examinations were normal. There was no history of epileptic seizures.

Patient B is a girl born at 38 + 4 weeks of gestation to first-cousin parents of Turkish origin (Family II; Fig 2A). Newborn screening indicated biotinidase deficiency (OMIM #253260), which was confirmed by enzymatic and molecular genetic analyses. Oral biotin supplementation was started immediately. Initially, low platelet counts were interpreted as secondary to known autoimmune thrombocytopenia of the mother and recovered spontaneously. Nevertheless, she had recurrent neutropenia (neutrophil counts $0.2–1.5 \times 10^9$/l, ref > $1.5 \times 10^9$/l) and repeatedly suffered from infections that required antibiotic treatment. Recurrent anemia was also noted (Hb 74–110 g/l). Psychomotor skills developed normally until 5 months, but from 7 months of age reduced movements and vocalizations were noted, together with progressive feeding difficulties. At 9 months of age, she could still reach out and grasp but not turn around or crawl and showed persisting primitive reflexes. Brain MRIs at 9 months (not shown) and 16 months showed decreased cerebral and cerebellar volumes and myelination had not significantly progressed between the investigations (Fig 1D). CSF protein was markedly elevated (7.138 g/l, ref 0.09–0.33 g/l). At 16 months of age, she had developed severe hypotonia and required gastric tube feeding. Dysmorphic facial appearance with a coarse face, a broad nasal bridge, and bushy eyebrows was obvious. Hepatomegaly was evident clinically and upon abdominal ultrasound. X-rays of chest, skull, and hand revealed dysostosis multiplex (Fig 1B and C), a feature of MPSs (Nicolas-Jilwan & AlSayed, 2018). At 21 months of age, the patient showed lack of voluntary movements and fixation and could not vocalize or communicate. An EEG was normal, and there was no history of epileptic seizures. At an age of 2 years and 3 months, the patient died following an episode of acute worsening and fever (details are unavailable).

### Clinical biochemistry and genetics

As the clinical phenotypes were highly suggestive of an LSD, both patients underwent extensive biochemical investigations. Routine

**Table 1. Summary of clinical phenotypes**

| | Patient A | Patient B | VPS16 c.2272-18C>A | VPS16 c.156C>A | VPS33A | VPS11 | VPS41 |
|---|---|---|---|---|---|---|---|
| **Clinical phenotypes** | | | | | | | |
| Coarse facial features, macroglossia, hypertrichosis | ✔ | ✔ | 2/2 | N/A | 15/16 | 0/15 | N/A |
| Developmental delay or regression | ✔ | ✔ | 2/2 | 0/5 | 16/16 | 15/15 | 3/3 |
| Pyramidal signs | ✔ | ✔ | 2/2 | N/A | 1/2 | 9/15 | 3/3 |
| Dysostosis multiplex | ✔ | ✔ | 2/2 | N/A | 16/16 | N/A | N/A |
| Joint contractures | ✔ | ✔ | 2/2 | N/A | 15/16 | 10/12 | N/A |
| Respiratory dysfunction | ✔ | ✔ | 2/2 | N/A | 16/16 | 2/15 | N/A |
| Renal dysfunction | – | – | 0/2 | N/A | 16/16 | 0/10 | N/A |
| Cardiomyopathy and/or heart failure | – | – | 0/2 | N/A | 12/16 | 0/10 | N/A |
| Hepatomegaly and/or splenomegaly | – | ✔ | 1/2 | N/A | 15/16 | 0/15 | N/A |
| Optic atrophy | ✔ | N/A | 1/1 | N/A | 5/15 | 3/11 | 1/1 |
| Anemia | ✔ | ✔ | 2/2 | N/A | 16/16 | 0/8 | N/A |
| Neutropenia or leukopenia | ✔ | ✔ | 2/2 | N/A | 9/16 | 0/8 | N/A |
| Thrombocytopenia | – | ✔ | 1/2 | N/A | 13/16 | 0/8 | N/A |
| Epilepsy | – | – | 0/2 | N/A | 0/15 | 10/15 | N/A |
| **Neuroradiology** | | | | | | | |
| White matter lesions and/or atrophy | ✔ | ✔ | 2/2 | 0/1 | N/A | 6/14 | 3/3 |
| Thin corpus callosum and/or hypomyelination | ✔ | ✔ | 2/2 | 0/1 | 3/5 | 14/14 | 3/3 |
| Decreased NAA on MR spectroscopy | ✔ | N/A | 1/1 | 0/1 | N/A | 1/1 | N/A |
| **Lab analyses** | | | | | | | |
| Urinary GAG | ✔ | – | 1/2 | N/A | 13/15 | N/A | N/A |
| Granulated lymphocytes | ✔ | N/A | 1/1 | N/A | 2/2 | N/A | N/A |
| | | | This paper | Cai et al, 2016 | Kondo et al, 2017 | Zhang et al, 2016 | Steel et al, 2020 |
| | | | | | Dursun et al, 2017 | Hörtnagel et al, 2016 | preprint: van der Welle et al, 2019 |
| | | | | | Pavlova et al, 2019 | Edvardson et al, 2015 | |

Summary of clinical phenotypes observed in patients, in comparison with other published cases resulting from recessive HOPS/CORVET subunit defects. N/A, data not available.

tests for inborn metabolic disorders, including initial urine GAGs (a marker for MPSs), plasma amino acids, acylcarnitine profile, and urine organic acids were normal. An extended screen in patient A found increased total sialic acid in urine (142 mmol/mol creatinine, ref 50–107 mmol/mol creatinine) with an abnormal pattern of acidic and neutral oligosaccharides. Chitotriosidase was also slightly elevated (61 nkat/l, ref < 40 nkat/l). However, enzymatic assays for known MPSs and other LSDs performed in both cases failed to identify the underlying cause (see supplementary material for details). Renewed analysis of urine GAGs when patient A was 3 years and 5 months old showed increased excretion (21 g/mol creatinine, ref 3–13 g/mol creatinine) and an abnormal pattern (65% chondroitin sulfate, 25% dermatan sulfate, and 10% heparan

sulfate), suggesting secondary effects in GAG catabolism with disease progression.

Patient A was first genetically investigated using a gene panel for known LSDs, including *PSAP* deficiency (see Table EV1A for included genes), which did not identify any pathogenic variants that could explain the patient's condition. As clinical suspicion of a recessive LSD remained, while metabolic tests were inconclusive, whole-genome sequencing was performed. Assessment of variants within the Lysoplex *in silico* gene panel (Di Fruscio *et al*, 2015; Table EV1C) identified a homozygous intronic variant (c.2272-18C>A) in *VPS16* (NM_022575.3) (Figs 2C and EV2A). Both parents were heterozygous carriers (Fig 2A). Patient B was investigated at 9 months of age using a gene panel for leukodystrophies

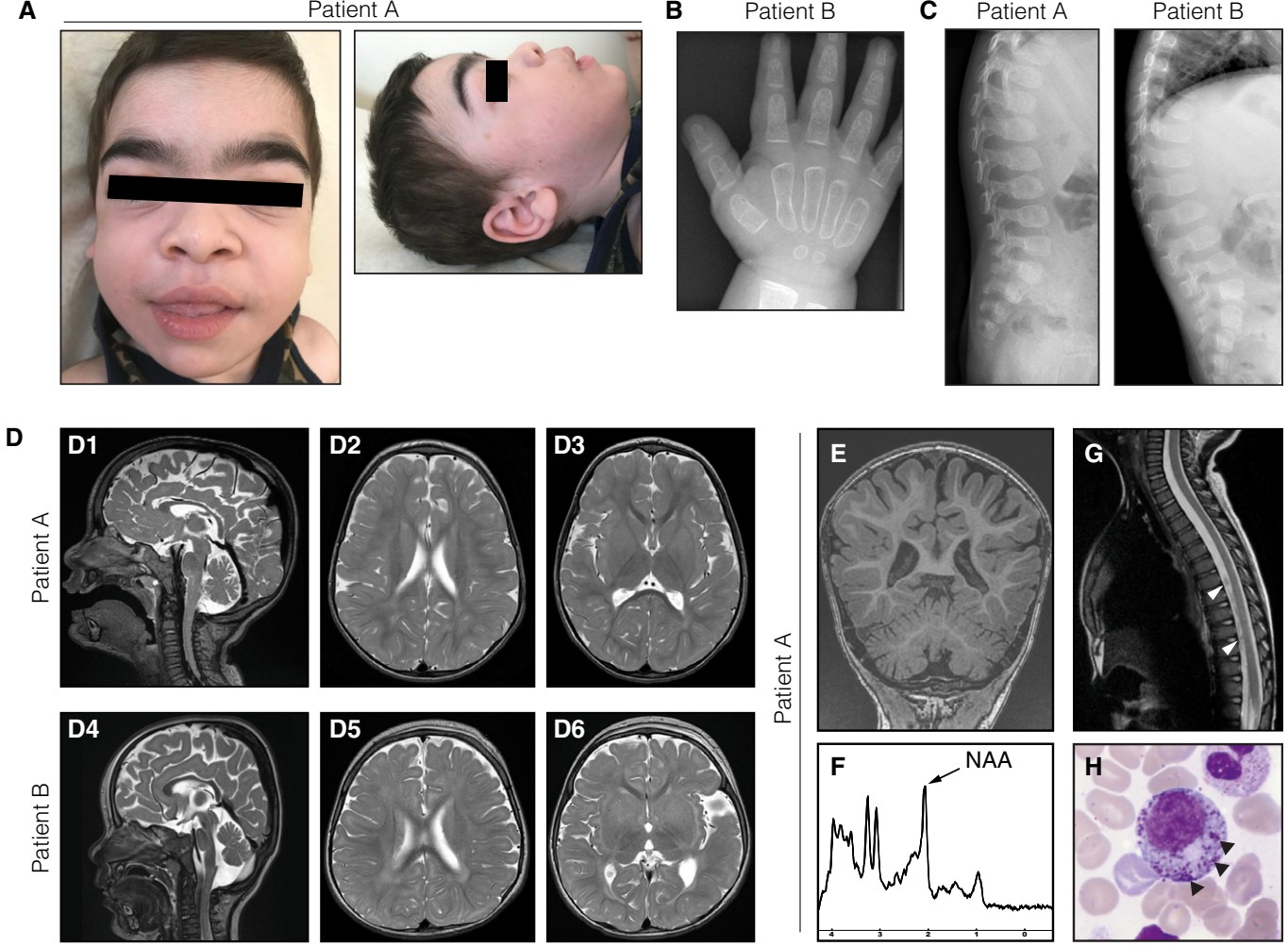

**Figure 1. Clinical features of the patients.**

A   Facial phenotypes of patient A at 3.5 years of age, showing coarse facial features, broad ear lobes, macroglossia, and hypertrichosis.

B   Radiograph of the hand of patient B at 21 months of age demonstrating rough trabecular structure and thin corticalis as signs of dysostosis multiplex.

C   Ovoid vertebrae visualized by spine radiographs of both patients.

D   T2-weighed brain MRIs of patient A (D1–3) at 28 months of age and patient B at (D4–6) at 16 months of age, showing global brain atrophy as well as periventricular and deep white matter hyperintensities.

E–G   Additional MR investigations of patient A at 28 months of age. (E) T1-weighed coronal images demonstrating thinning of the posterior corpus callosum and reduced volumes of the cerebellar hemispheres and vermis. (F) Magnetic resonance spectroscopy of the white matter showing a decreased N-acetylaspartate (NAA) peak at 2.02 ppm (arrow). (G) T1-weighed MRI of the spinal cord showing contrast enhancements around the conus medullaris (arrowheads).

H   May Grunwald-Giemsa-stained bone marrow smear from patient A with myelopoietic cells containing dense granules (arrowheads), suggestive of a lysosomal storage disorder.

(Table EV1B), which revealed a homozygous *BTD* variant responsible for the biotinidase deficiency (c.476G>A, p.Arg159His) as well as a homozygous variant of unknown significance in *SCN2A* (c.5995G>T, p.Asp1999Tyr), which was not interpreted as being causative (see Discussion). As the clinical phenotype at that point clearly suggested a LSD, trio exome sequencing was performed at 16 months of age and the identified variants were filtered using the Lysoplex *in silico* gene panel (Di Fruscio *et al*, 2015; see Table EV1C for included genes). This approach only identified the same c.2272-18C>A *VPS16* variant, for which the patient was homozygous and both parents were heterozygous carriers (Fig 2B). Thus, the patients were diagnosed independently and subsequently linked by means of

GeneMatcher (Sobreira *et al*, 2015). There is no established ancestral link between the families, although this seems likely since both report some roots in Turkey. VPS16:c.2272-18C>A is not observed in the GnomAD database (v. 2.1.1) (Karczewski *et al*, 2020), although it may be present in Middle Eastern populations that are underrepresented in this dataset.

## The *VPS16*:c.2272-18C>A variant disrupts normal splicing and generates a hypomorphic allele

The c.2272-18C>A variant is predicted to create a splice-acceptor site 16 bp upstream of exon 23 (Fig EV2B). To confirm this, we

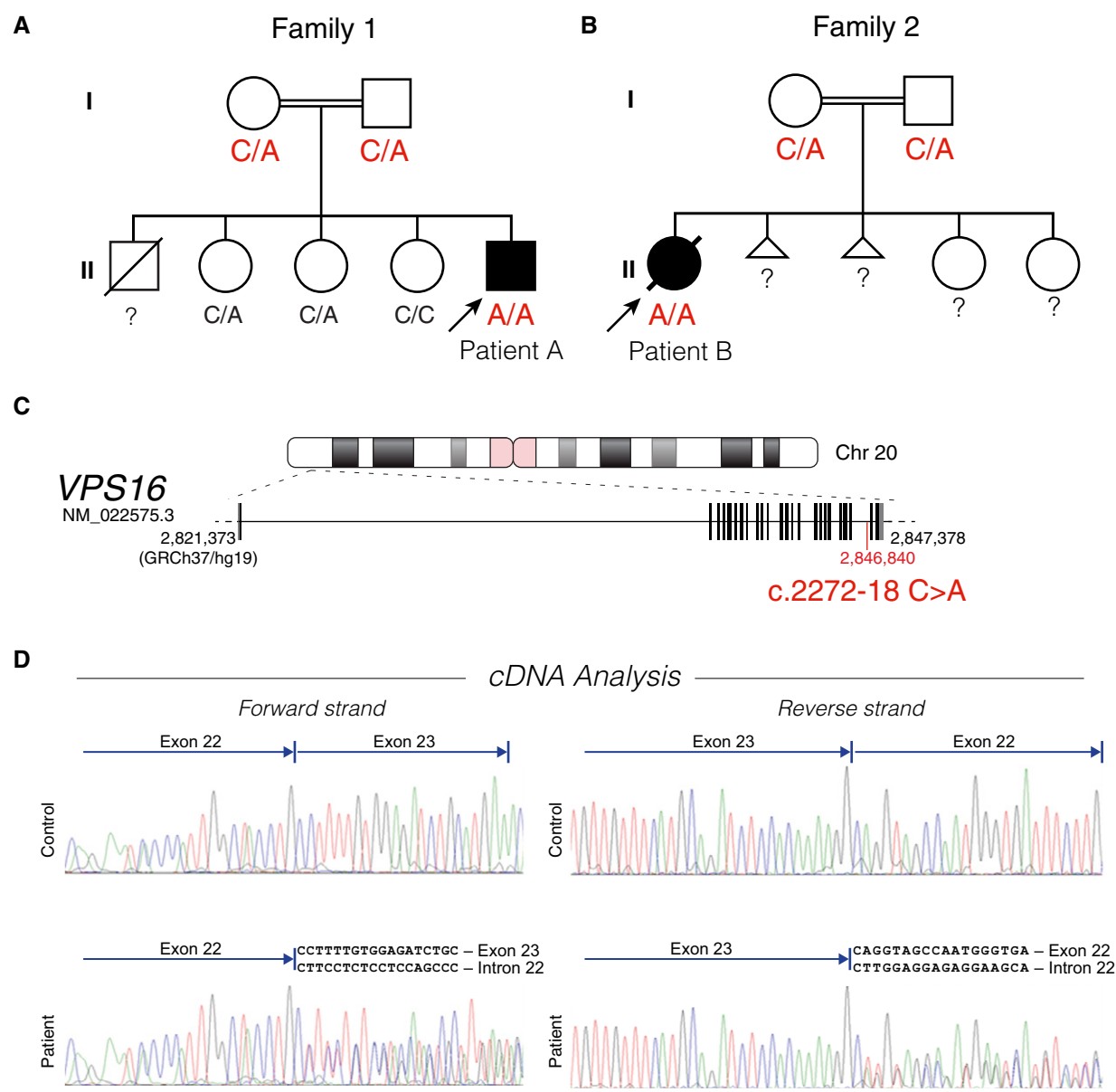

**Figure 2. Family pedigrees and mutation analysis.**

A, B    Pedigrees of the two families, indicating autosomal recessive inheritance. The genotype of the *VPS16* c.2272-18 position is indicated.?, not available for testing. (A) Family of patient A. The three sisters are healthy, but an older brother had previously died at 11 months from suspected heart failure after showing signs of developmental delay from an age of 3 months of age. (B) Family of patient B. The mother previously had two spontaneous miscarriages but the two sisters are healthy.

C    Position of the intronic c.2272-18 variant in the VPS16 gene.

D    cDNA analysis by Sanger sequencing of leukocyte RNA from patient A and a control sample. Reads beyond the alternatively spliced breakpoint yield double reads in patient samples, corresponding to the predicted mis-splicing and wild-type sequences.

analyzed cDNA isolated from blood leukocytes. Sanger sequencing of patient cDNA showed mixed reads beyond the breakpoint that corresponded to both the predicted mis-spliced sequence as well as normally spliced sequence (Fig 2D). The mis-spliced transcript retains part of intron 22, which introduces a frameshift and a premature stop codon (Fig 3A).

To further assess the functional consequences of the c.2272-18C>A variant, we established skin-derived fibroblasts from both

patients. Quantitative RT–PCR with a probe spanning the mis-spliced intron-exon border revealed that wild-type *VPS16* transcripts were reduced by ~ 85% in patient-derived cells as compared to fibroblasts derived from control subjects (Fig 3B). Total levels of *VPS16* transcripts were also reduced, compatible with nonsense-mediated decay of mis-spliced transcripts (Fig 3B). Quantitative immunoblotting using an antibody raised against the N-terminal part of VPS16 revealed a similar ~ 85% reduction in full-length

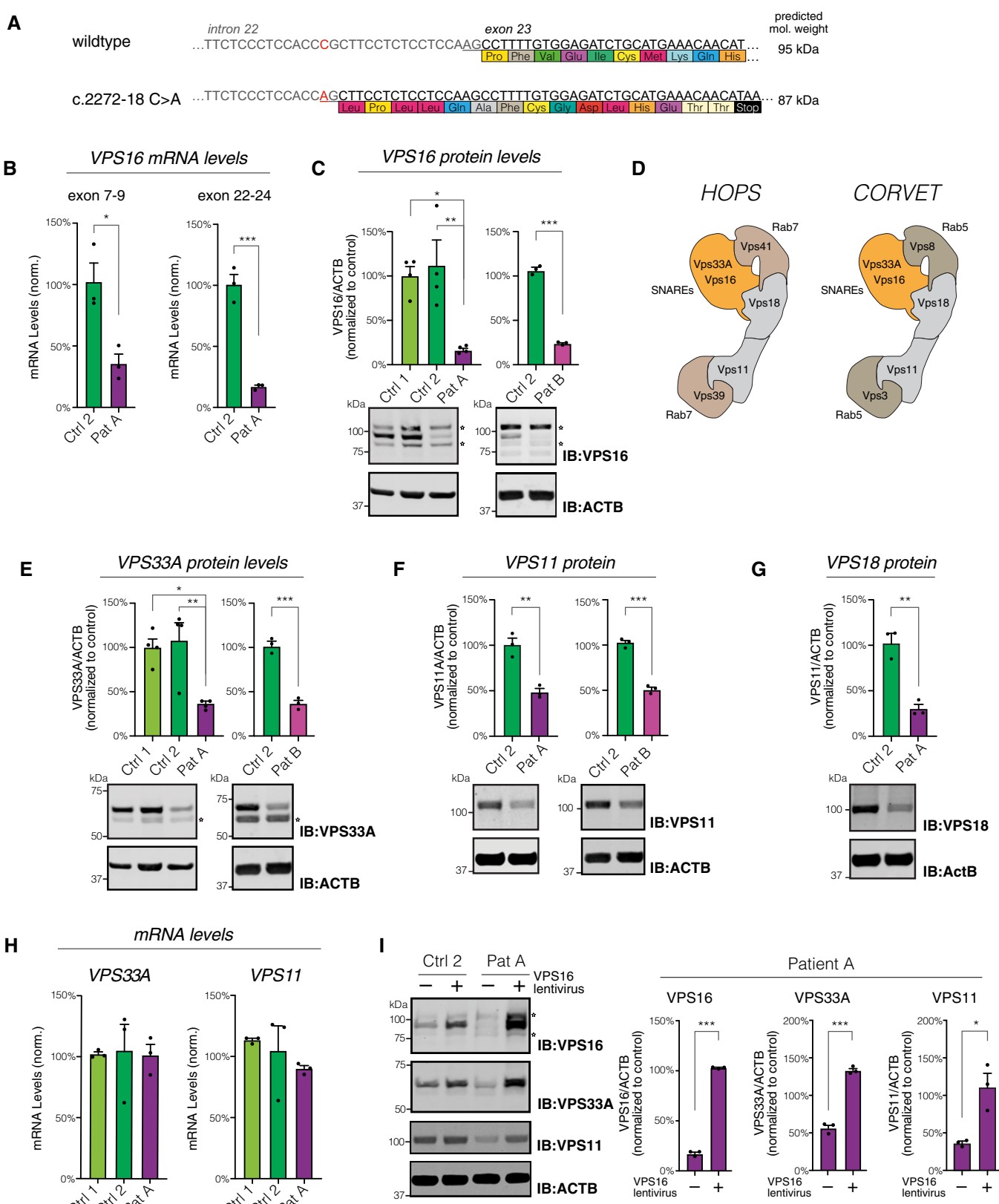

**Figure 3.**

**Figure 3. The c.2272-18C>A mutation impairs VPS16 splicing and leads to reduced levels of VPS16 and HOPS/CORVET components.**

A   The c.2272-18C>A variant generates a new splice-acceptor sequence 16 base pairs upstream of the normal exon 23 border, resulting in a frameshift and premature stop codon.

B   Levels of VPS16 transcripts in fibroblasts by qPCR. The probe of the exon 22–24 PCR spans the mis-spliced intron-exon border. Data were normalized to levels of POLR2A using the ΔΔCt method.

C   Quantifications of VPS16 in fibroblast lysates by immunoblotting. Representative immunoblots (bottom) and summary quantifications (top) of the levels of VPS16 in patient A (n = 4) and patient B (n = 3), normalized to levels of actin (ACTB) and expressed as % of controls.

D   Schematic illustration of HOPS and CORVET complexes showing their subunit compositions. The Rab5-binding CORVET subunits VPS3 and VPS8 replace the Rab7-binding VPS39 and VPS41 of HOPS. Models adapted from Bröcker et al (2012).

E–G  Quantifications of HOPS/CORVET subunits in fibroblast lysates (top) and representative immunoblots (bottom), analyzed as in (C). (E) VPS33A, (F) VPS11, (G) VPS18.

H   Analysis of VPS33A and VPS11 mRNA levels in fibroblasts by qPCR. Data were normalized to levels of POLR2A using the ΔΔCt method.

I   Immunoblots (left) and quantifications normalized to control (right) of VPS16, VPS33A, and VPS11 in lysates from fibroblasts transduced with a lentivirus to express VPS16 or control (empty vector).

Data information: All immunoblots were acquired using fluorescent secondaries and pseudocolored to greyscale. Stars denote unspecific bands. All data were shown as mean ±SEM of 3–4 biological replicates. Statistical comparisons were made by two-tailed unpaired Student's t-tests [panels B, C (right), E (right), F, G and I] or one-way ANOVA with Dunnett's multiple comparisons tests [panels C (left) and E(left)]. *P < 0.05; **P < 0.01; ***P < 0.001.

protein in cells from both patients (Fig 3C). No truncated VPS16 protein could be detected, suggesting that the mis-spliced protein is rapidly degraded. To further test this possibility, we treated the cells with the proteasome inhibitor MG132 for up to 2 h. Proteasome inhibition caused a slight increase in VPS16 and VPS33A levels in controls, but no truncated protein with the predicted molecular weight of 87 kDa (wild-type protein is 95 kDa) was detected (Fig EV3A and B). Thus, while the mutation causes mis-splicing of most VPS16 transcripts, some transcripts escape the new splice-acceptor site to generate wild-type transcripts that are sufficient to encode for ~ 15% residual wild-type protein.

## Impaired VPS16 expression reduces cellular levels of HOPS/CORVET subunits

In order to determine the functional consequences of reduced levels of VPS16, we assessed the levels of other HOPS/CORVET subunits, given the central role of VPS16 in each of these complexes (Fig 3D). VPS33A, a direct binding partner of VPS16, was found to be reduced by ~ 70% in patient-derived fibroblasts in comparison with control cells (Fig 3E). Levels of VPS11 and VPS18, two other core HOPS/CORVET subunits, were also reduced to a similar extent in patient-derived cells (Fig 3F and G). This reduction was not a consequence of transcriptional regulation, as both VPS11 expression and VPS33A expression were comparable to that of control fibroblasts (Fig 3H), but suggested that loss of VPS16 destabilized other HOPS/CORVET subunits. To directly test this, we generated lentiviruses that express full-length human VPS16. We used this virus or control (empty vector) to transduce fibroblasts from patients and controls and again assessed the protein levels of HOPS/CORVET subunits (Fig 3I). Reconstituting VPS16 expression in patient-derived cells significantly increased the levels of both VPS33A and VPS11 subunits (Fig 3I), whereas its overexpression had only limited effects in control cells. These results suggest that reduced levels of VPS16 directly cause the observed reduction in cellular HOPS/CORVET components. We additionally tested whether VPS16 with the N52K missense mutation, previously reported in primary dystonia (Cai et al, 2016), could rescue VPS11 and VPS33A levels. Lentiviral expression of VPS16[N52K] could complement the patient phenotype by normalizing cellular VPS33A and VPS11 levels, similar to wild-type VPS16 (Fig EV3C and D).

## Impaired transferrin uptake and sorting in patient-derived fibroblasts

To investigate the cellular consequences of the VPS16 mutation, we analyzed endosomal compartments in patient-derived fibroblasts. First, we assessed early endosomes by staining fibroblasts from patient and control subjects, transduced with VPS16-expressing or control lentiviruses, for the early endosome marker EEA1 (Fig 4A). There were no differences in the sizes or fluorescence intensities of EEA1-positive puncta (Fig 4B). To investigate traffic through endocytic pathways, we assessed transferrin uptake (Harding et al, 1983; Maxfield & McGraw, 2004). Fibroblasts from patient and control subjects were transduced with lentiviruses to express either wild-type VPS16, N52K-mutant VPS16, or empty vector. After 2 h of serum starvation to remove competing transferrin in the media, the cells were fed with fluorescently labeled transferrin for 30 min, followed by an acidic wash to remove surface-bound, but not endocytosed, protein. Analysis by confocal microscopy revealed a defect in the transferrin uptake of patient-derived fibroblasts (Fig 4C). Quantitative assessment revealed a decrease in the amounts of fluorescent puncta (Fig 4D) and fluorescence intensities (Fig 4D), compared with either control cells or patient cells complemented by re-expression of VPS16. VPS16[N52K] rescued the phenotype similarly to the wild-type protein (Fig 4C and D). Absolute quantification of fluorescent transferrin uptake in cellular lysates revealed a ~ 25% decrease in patient cells (Fig EV4A).

In order to assess whether the impaired transferrin uptake could be explained by a decrease in transferrin receptor levels, we analyzed transferrin receptor levels by quantitative immunoblotting. The levels of transferrin receptors were instead found to be increased by ~ 30% in patient cells (Fig 4E and F). The increase was similar under both serum-starved and non-starved conditions (Fig 4E) and normalized upon re-expression of VPS16 (Fig 4F). This finding may reflect an accumulation of transferrin receptors in endosomal compartments. As Rab11, a small GTPase of recycling endosomes, has previously been shown to mediate transferrin receptor recycling, and its depletion caused intracellular accumulation of transferrin receptors (Ren et al, 1998; Takahashi et al, 2012), we measured levels of Rab11 by immunoblotting. We found no differences in the levels of Rab11 between patient and control cells (Fig EV4B). To further study the fate of internalized transferrin receptors, we stained cells for

transferrin receptors but were unable to obtain a sufficiently specific signal. We instead assessed the subcellular distribution of endocytosed transferrin, as this remains receptor-bound until recycled to the cell surface (Maxfield & McGraw, 2004). We fed cells with fluorescent transferrin and simultaneously loaded cells with LysoTracker dye to stain acidic compartments and analyzed their co-localization

by confocal microscopy (Fig 4G). Significantly more of the endocytosed transferrin co-localized with LysoTracker signal in patient-derived cells when compared to control cells or cells rescued by VPS16 re-expression. To corroborate this finding, we also stained cells for LAMP2 and analyzed its co-localization with endocytosed transferrin. The results were similar to those obtained with

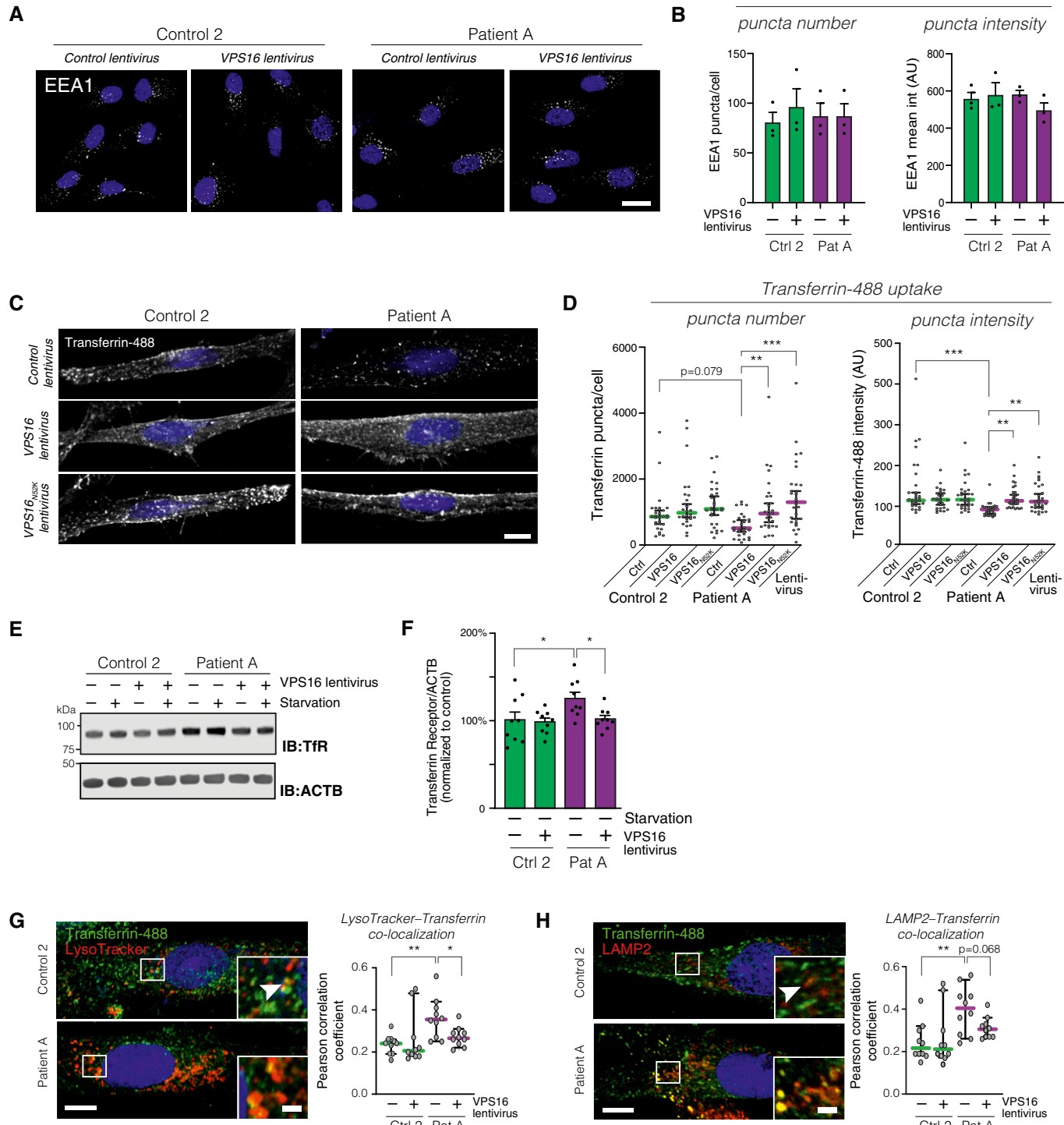

Figure 4.

**Figure 4.  Analysis of transferrin uptake and trafficking in fibroblasts.**

A, B    Analysis of the early endosome marker EEA1 in fibroblasts transduced with control or *VPS16*-expressing lentivirus by confocal microscopy. (A) Representative confocal micrographs. Scale bar, 20 μm. (B) Quantification of EEA1-stained puncta number and intensities (ca 80 cells analyzed for each of *n* = 3 biological replicates).

C       Analysis of transferrin uptake in fibroblasts transduced with the indicated lentiviruses. Representative confocal micrographs of fibroblasts fed with Alexa488-conjugated Transferrin for 30 min. Scale bar, 10 μm.

D       Quantifications of the number and fluorescence intensities of intracellular Transferrin puncta (*n* = 28–30 fields from three independent experiments). Colored horizontal bars indicate median values and whiskers 5 and 95 percentiles.

E       Immunoblot for Transferrin receptor (TfR) levels in the indicated fibroblasts under basal and serum-starved conditions.

F       Summary quantification of Transferrin receptor levels in non-starved fibroblasts transduced with the indicated lentiviruses, normalized to actin (ACTB), and expressed as % of controls (*n* = 6 biological replicates).

G, H    Analysis of intracellular Transferrin trafficking. Fibroblasts fed with fluorescently labeled Transferrin were co-stained with LysoTracker (G) or an antibody against LAMP2 (H). Representative images (left) of fibroblasts transduced with control virus, with magnified areas demonstrating co-localized puncta (arrowheads). Scale bars 10 μm or 2 μm (inserts). Quantifications (right) of the co-localization between Transferrin and LysoTracker or LAMP2, respectively, expressed as Pearson correlation coefficients. Colored horizontal bars indicate the median values and whiskers 5 and 95 percentiles (*n* = 10 optical sections from three independent experiments).

Data information: Bar graphs represent data as mean ±SEM. Statistical comparisons between the indicated groups by one-way ANOVA with Holm–Sidak's multiple comparisons tests. *$P$ < 0.05; **$P$ < 0.01; ***$P$ < 0.001.

LysoTracker (Fig 4H). Thus, our results indicate that the reduction in HOPS/CORVET complexes caused by the mutation in VPS16 leads to reduced transferrin uptake and a relative accumulation of transferrin/receptor complexes in acidic late endosomal/lysosomal compartments or a recycling-late endosome hybrid organelle.

### Accumulation of acidic compartments in patient cells

Next, we directly assessed acidic compartments in patient-derived and control fibroblasts by labeling serum-starved cells with Lyso-Tracker (Fig 5A). Patient-derived cells showed a significant accumulation of LysoTracker-stained puncta, reflected by increased number and signal intensity of the labeled puncta (Fig 5B). Lentiviral expression of *VPS16* restored the LysoTracker phenotype to that of control cells (Fig 5B). To test whether the LysoTracker-stained puncta corresponded to late endosomes and lysosomes, we co-stained cells for the marker lysosome-associated membrane protein 2 (LAMP2) (Fig EV4C). LAMP2-positive puncta co-localized with LysoTracker to a similar extent in both patient and control cells (Fig EV4D). However, we could not detect a corresponding increase in the number or signal intensities of LAMP2-stained puncta in patient cells (Fig EV4E), possibly due to lower signal-to-noise of the antibody staining (Fig EV4C). We then quantified protein levels of LAMP1 and LAMP2 to test whether the expansion of LysoTracker-stained compartments was associated with an increase in lysosomal markers. There was no detectable increase in levels of LAMP1 (Fig 5C) or LAMP2 (Fig 5D). As a measure of lysosomal function, we assessed processing of Cathepsin D, which is subject to proteolytic processing in lysosomes to generate mature forms (Richo & Conner, 1991). Cathepsin D processing was slightly, but not significantly, reduced in patient-derived cells (Fig 5E). The above findings show that patient-derived cells contain an accumulation of acidic compartments but suggest that similar amounts of lysosomal proteins are distributed to this expanded pool of vesicles.

### Accumulation of autophagosomes but normal flux in patient-derived fibroblasts

As the HOPS complex additionally mediates fusion between lysosomes and autophagosomes (Jiang *et al,* 2014; Wartosch *et al,* 2015),

we assessed autophagosome formation and turnover in patient-derived cells. First, we measured the levels of the autophagy receptor p62/SQSTM1 (Fig 6A), which is known to decrease upon induction of autophagy and to accumulate when autophagy is blocked, at least in some cell types (Klionsky *et al,* 2016). We saw no significant changes in the levels of p62, although it was slightly higher in patient cells and tended to decrease as expected in control cells upon serum starvation (Fig 6B). Next, we studied the conversion of LC3 isoforms during autophagy by immunoblotting (Fig 6C). As LC3-II was below reliably quantifiable levels under basal conditions in our fibroblasts, we stimulated autophagy by subjecting the cells to 2 h of serum starvation prior to analysis. Levels of LC3-II, normalized to LC3-I to reduce sample-to-sample variability, were significantly increased in patient-derived fibroblasts (Fig 6D; see Appendix Fig S1 for levels of each isoform); re-expression of *VPS16* normalized this difference. Blocking autophagosome–lysosome fusion with the inhibitor Bafilomycin A1 caused a similar relative increase in both control and patient cells (Fig 6E), indicating that the total autophagic flux was normal. Moreover, we quantified autophagosomes in fibroblasts by analyzing LC3 puncta in cells expressing a tandem mRFP-GFP-LC3 reporter construct (Kimura *et al,* 2014). Combined green and red LC3 puncta, which reflect non-acidified autophagosomes, were quantified by live confocal microscopy. This revealed an increased number of autophagosomes in patient cells under basal conditions (Fig 6F and G). Starvation induced autophagosome formation in control cells, as expected, but not in patient-derived cells, which instead showed a paradoxical decrease. The numbers of red-only LC3-puncta, which represent acidified autolysosomes, normalized to the total number of red LC3-puncta were decreased in patient-derived cells under both basal and starved conditions (Fig 6H). This result may reflect a decreased rate of autophagosome-lysosome fusion, consistent with the previously reported role for VPS16 in this process (Wartosch *et al,* 2015), but could also reflect impaired acidification of autolysosomes. Labeling with LysoTracker revealed that stained puncta almost perfectly overlapped with red-only LC3-containing autolysosomes, in which the GFP signal is quenched by the acidic environment (Fig 6H). This demonstrated the specificity of the reporter construct and excluded a major defect in the autolysosome acidification. Moreover, it confirmed that the patient-derived cells maintain capacity to form autolysosomes.

**A**

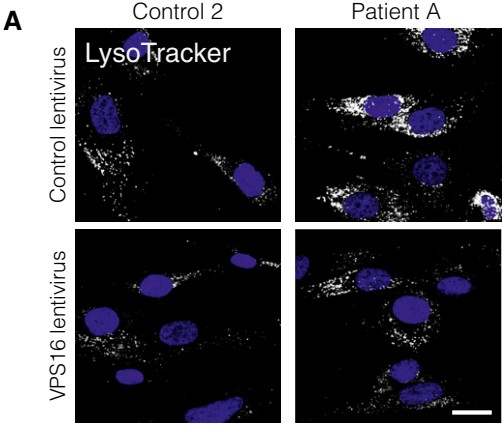

**B**

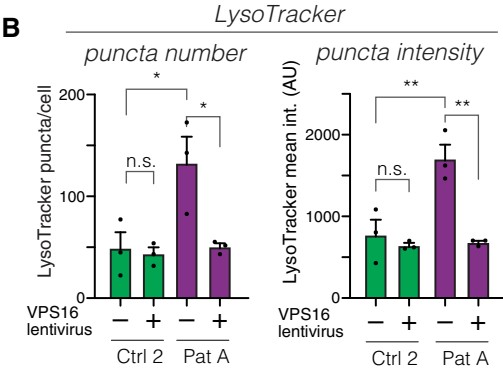

**C**

**D**

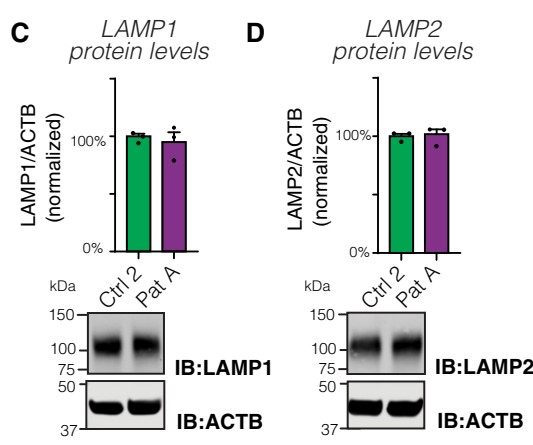

**E**

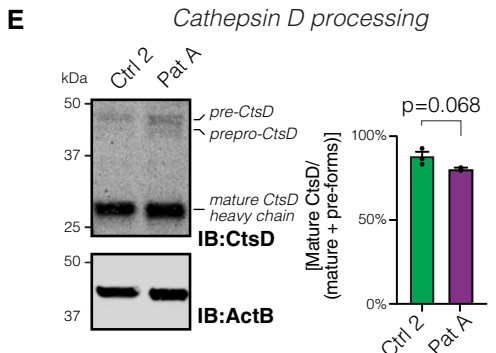

Figure 5. Analysis of lysosomal distribution and function in fibroblasts.

A Confocal micrographs of fibroblasts transduced with control or *VPS16*-expressing lentivirus and labeled with LysoTracker. Scale bar, 20 μm.
B Quantification of LysoTracker-stained puncta number and intensities (ca 80 cells analyzed for each of *n* = 3 biological replicates).
C, D Representative immunoblots (bottom) and summary quantifications (top) of the levels of LAMP1 (C) and LAMP2 (D) in fibroblasts, normalized to levels of actin (ACTB) and expressed as % of controls (*n* = 3 biological replicates).
E Analysis of lysosomal Cathepsin D (CtsD) processing by immunoblotting. Immunoblot to detect CtsD isoforms (left) and quantification (right) of processing, calculated as the percentage of CtsD divided by the sum of all isoforms (*n* = 3 biological replicates).

Data information: Bar graphs represent data as mean ±SEM. Statistical comparisons between the indicated groups by two-tailed unpaired Student's *t*-tests (panel E) or one-way ANOVA with Holm–Sidak's multiple comparisons tests (panel B). *$P < 0.05$; **$P < 0.01$.

Taken together, these results demonstrate that the patient-derived fibroblasts have an increased number of autophagosomes, which may result from a decreased autophagosome–lysosome fusion rate.

### Loss of *ups16* causes impaired myelination in zebrafish

While the patient-derived fibroblasts show a clear cellular phenotype, the clinical picture is characterized by defects in the brain. To study how the cellular defects that arise from *VPS16* deficiency affect brain development, we generated a *vps16* zebrafish model. Zebrafish have a single homologous *vps16* sharing 68% amino acid identity with the human protein (Figs EV5A and 7A). Zebrafish larvae were made deficient for *vps16* by injecting Cas9 protein complexed with a guide RNA (gRNA) as previously described (Kuil *et al*, 2019). This consistently produced highly efficient indels and disruption of *vps16* (Fig EV5B and C), which lead to a consequent lack of swim bladder development and a clear pigmentation phenotype that allowed us to readily identify mutagenized embryos (crispants) (Fig 7B). Reduced melanin in the skin and retinal pigment epithelium is also observed in mutants for other HOPS components (Maldonado *et al,* 2006; Schonthaler *et al,* 2008; Thomas *et al,* 2011) and, along with the lack of a swim bladder, may reflect altered development of the lysosome-related organelles responsible for melanin and surfactant production, respectively.

As our patients presented with myelination abnormalities, we assessed whether disruption of *vps16* affects myelination during brain development. To this end, we disrupted *vps16* in transgenic zebrafish expressing membrane-tethered GFP under control of the myelin-binding protein (*mbp*) promoter. Crispants revealed fewer GFP-labeled myelin sheets, indicating a significantly lower level of myelination in the mid- and hindbrain regions at 5 days post-fertilization (dpf) (Fig 7C and D).

### Accumulation of acidic compartments and autophagosomes in *ups16* crispant brains

We assessed the number and distribution of acidic, lysosomal compartments in *vps16* crispants by *in vivo* LysoTracker imaging.

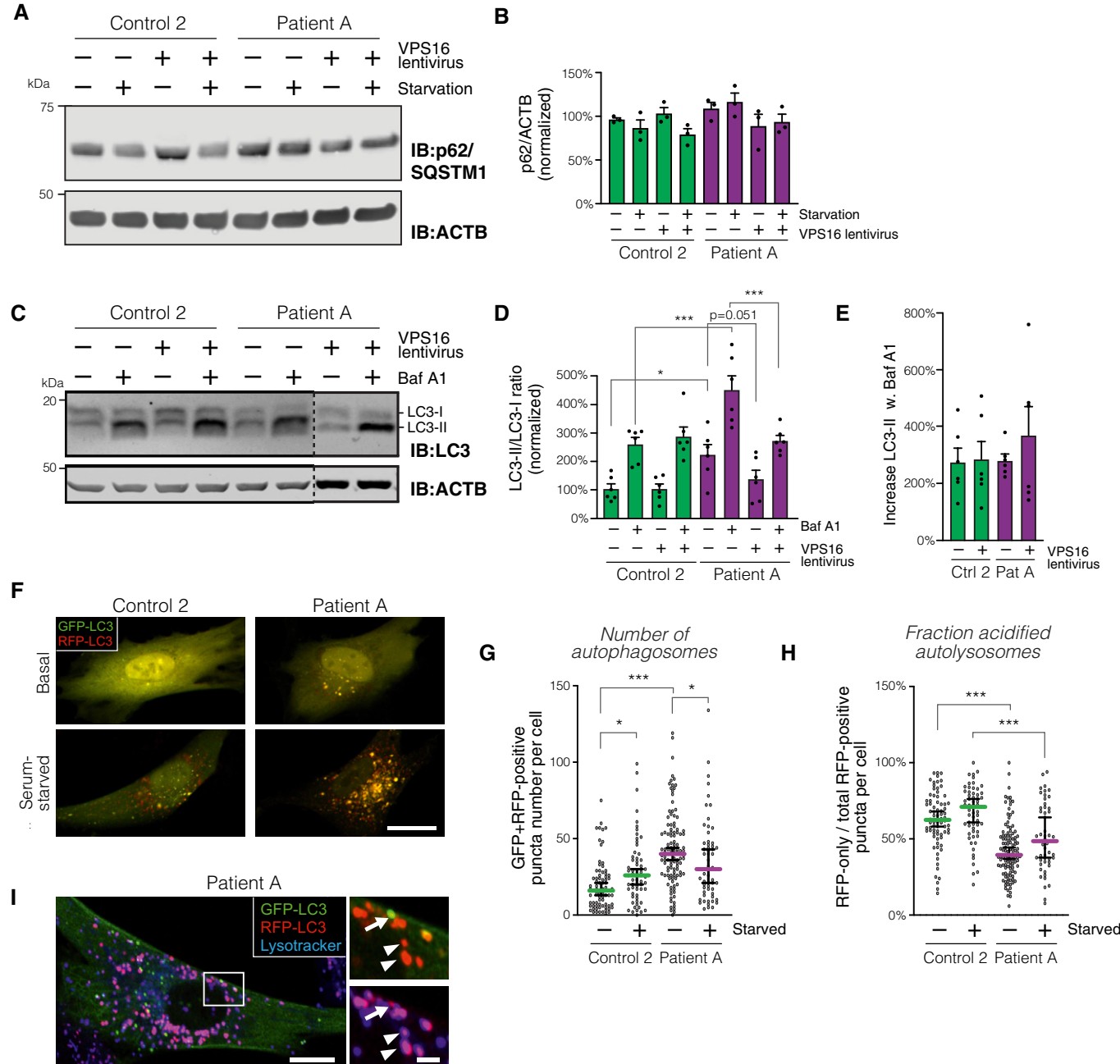

**Figure 6. Analysis of autophagosome formation and turnover in fibroblasts.**

A  Analysis of p62/SQSTM1 levels in patient and control fibroblasts transduced with control or *VPS16*-expressing lentivirus under basal conditions and after 2 h of serum starvation.

B  Summary quantification of (A), normalized to actin (ACTB) and expressed as % of controls (*n* = 3 biological replicates).

C  Analysis of LC3 isoforms in serum-starved fibroblasts transduced with control or *VPS16*-expressing lentivirus and treated with Bafilomycin A1 (Baf A1), as indicated.

D  Quantification of LC3-II, normalized to levels of LC3-I, actin and expressed as % of controls (*n* = 6 biological replicates).

E  Autophagic flux, calculated as the relative increase in LC3-II ratios upon addition of Baf A1 (data from D; *n* = 6 biological replicates).

F–I  Analysis of autophagosome numbers in fibroblasts transfected to express mRFP-GFP-LC3. (F) Representative confocal micrographs (optical sections) of fibroblasts imaged live under basal or serum-starved conditions. Scale bar, 20 μm. (G and H) Quantification of autophagosomes, defined as green + red puncta (G) and autophago-lysosomes, calculated as the number of red (but not green) puncta divided by the total number of puncta. Colored horizontal bars denote median values and whiskers 5 and 95 percentiles (*n* = 50–113 cells from a total of three independent experiments; for exact values see Appendix Table S1). (I) Cell of patient A transfected with mRFP-GFP-LC3, labeled with LysoTracker, and imaged live by confocal microscopy under serum-starved conditions. Insert shows red+blue autolysosomes (arrowheads) and a green+red autophagosome devoid of LysoTracker signal (arrow). Scale bars 10 μm or 2 μm (inserts).

Data information: Bar graphs represent data as mean ±SEM. Statistical comparisons between the indicated groups by one-way ANOVA with Holm–Sidak's multiple comparisons tests. **P* < 0.05; ****P* < 0.001.

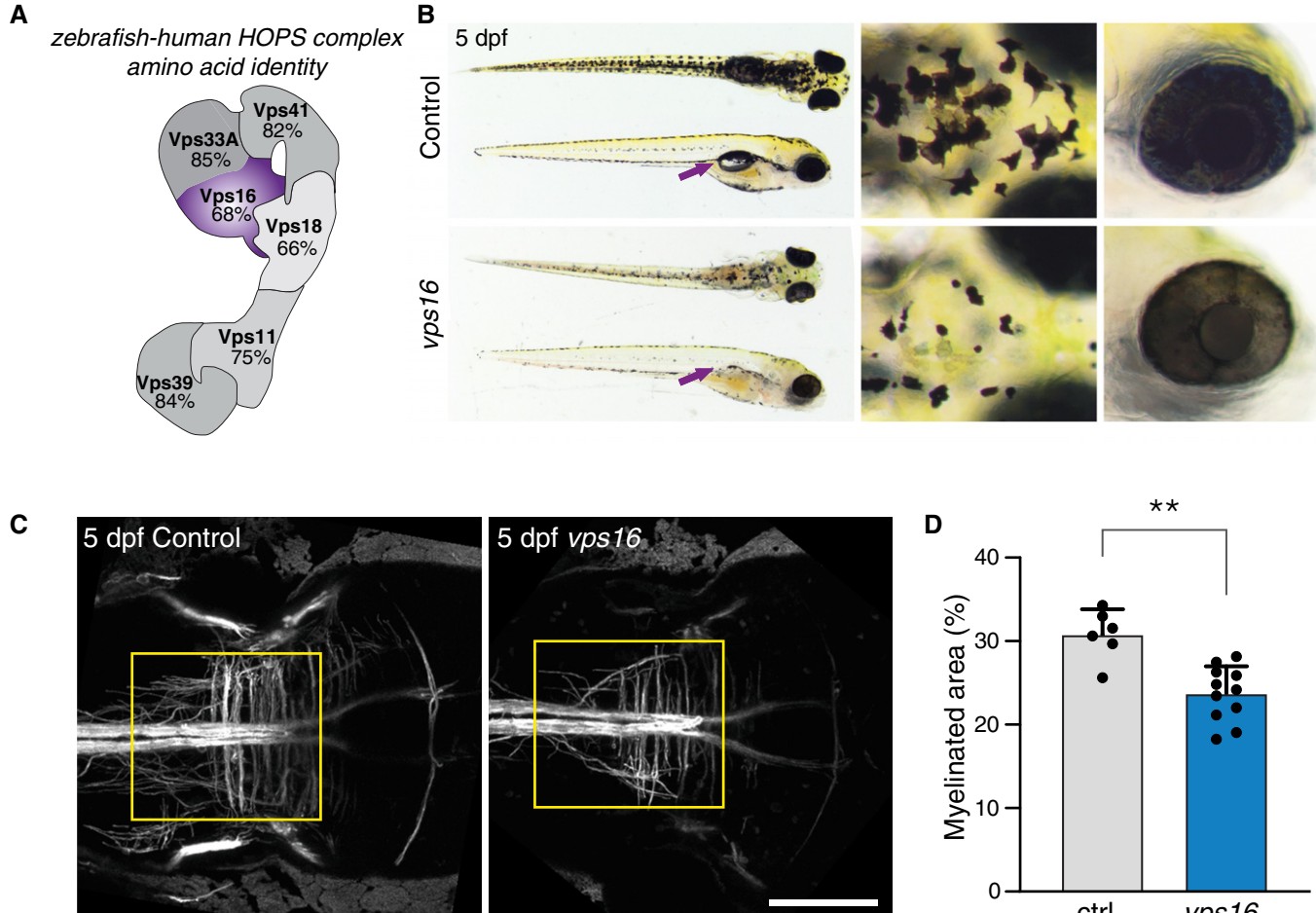

**Figure 7. Zebrafish embryos deficient in *vps16* demonstrate abnormalities in LROs and myelination.**

A   Schematic representation of the HOPS complex, indicating the high amino acid identity between human and zebrafish for each subunit.
B   Comparison of control and *vps16* embryos at 5 dpf. Compared with control, *vps16* embryos have reduced pigmentation in the skin and retinal pigment epithelium and do not develop a swim bladder (arrows). Magnified views show altered melanocyte morphology on the head and pale, translucent eyes.
C, D   Myelination, as reported using the *Tg(mbp:gfp-caax)* transgenic, is significantly reduced in *vps16* crispants. The yellow box indicates the region assessed in (D). Scale bar 100 μm. Bar graphs represent data as mean ±SEM. Statistical comparisons by Mann–Whitney *U*-tests (*n* = 6). **P < 0.01.

LysoTracker-stained puncta were found predominantly in the brain, with similar average signal intensities across control and *vps16* crispants at 3 dpf (Fig 8A and B). However, by 5 dpf LysoTracker signal was strongly increased throughout the brains of *vps16* crispants (Fig 8A and B), coinciding with the impaired myelination.

Next, we employed transgenic zebrafish expressing fluorescent proteins under the pan-neuronal *elavl3* (Ahrens *et al*, 2013) or astroglial *slc1a2b* (Kuil *et al*, 2019) promoters to further characterize the LysoTracker phenotype seen in the *vps16* embryos. This revealed that LysoTracker signal largely co-localized with astroglia, with little apparent overlap with neurons (Fig 8C), suggesting predominant early glial dysfunction. A closer look revealed large, LysoTracker-bright puncta in the crispant brains at both 3 and 5 dpf in a pattern reminiscent of the distribution of microglia at these stages. To further examine this phenotype, we examined transgenic zebrafish expressing GFP under the *mpeg1* promoter to label microglia (Ellett *et al*, 2011). This showed a dramatic increase in the number of

LysoTracker-stained acidic compartments in microglia (Fig 8D and E), in agreement with the increased accumulation of acidic compartments observed in patient-derived fibroblasts (Fig 5A and B) and myelopoietic cells (Fig 1H). Labeled microglia also showed a substantially altered, raspberry-like morphology. This phenotype is reminiscent of, yet more severe than, the previously described microglial phenotype of zebrafish larvae deficient in *hexb*, which is involved in another LSD (Kuil *et al*, 2019).

To assess autophagosomes in the brains of our crispant embryos, we used a transgenic zebrafish line that expresses GFP-tagged Lc3 protein downstream of the human cytomegalovirus (CMV) promoter (He *et al*, 2014). We observed that *vps16* embryos, in the absence of any stimulation, demonstrated strong increases in GFP-Lc3 signal throughout the brain as compared to controls (Fig 8F). Closer imaging revealed an increase in overall cytoplasmic signal in the crispants, with clusters of GFP-positive puncta found predominantly along the midline of the optic tectum and in the anterior hindbrain.

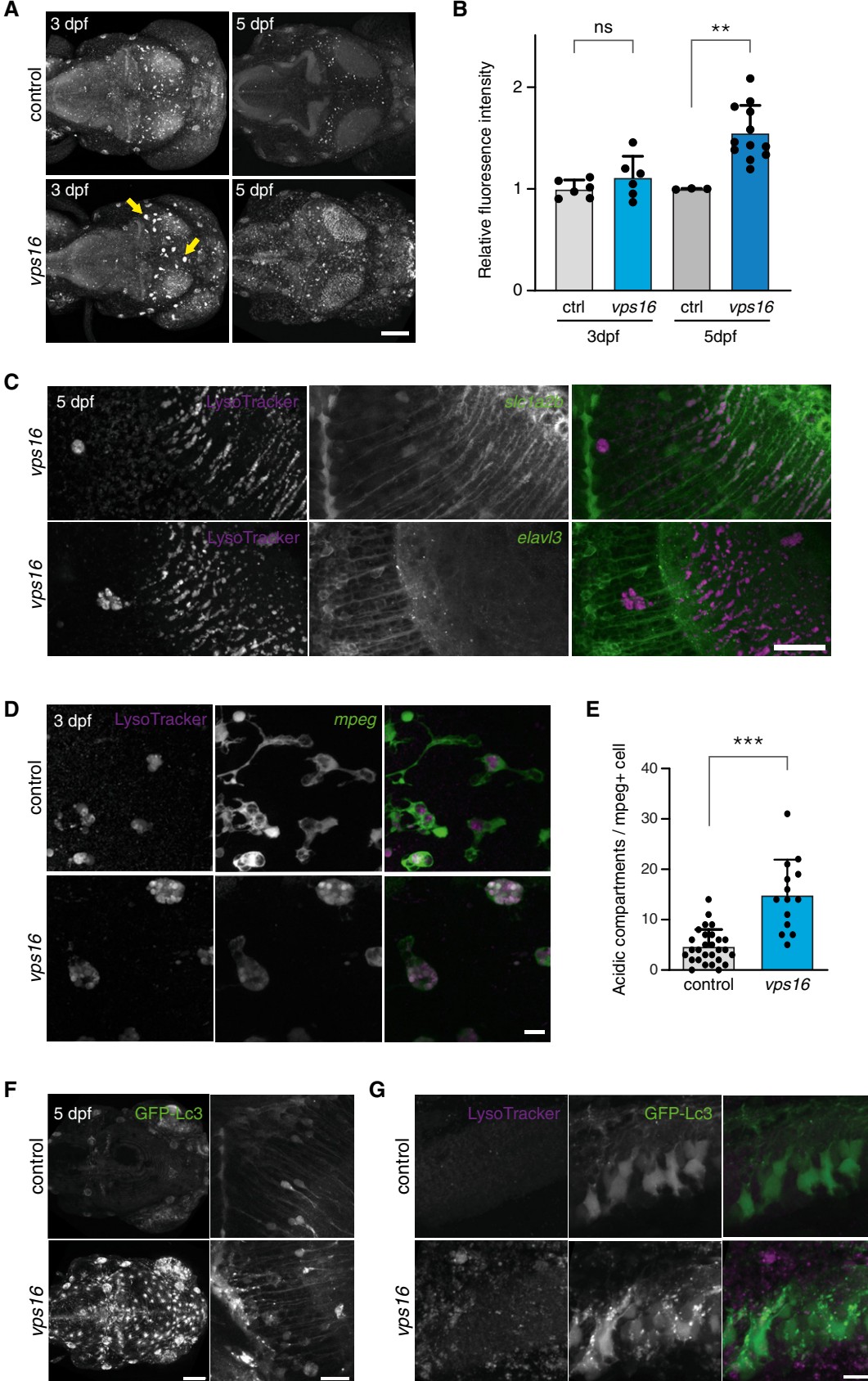

**Figure 8.**

**Figure 8. LysoTracker and GFP-Lc3 signal are enhanced in *ups16* embryos, suggesting glial dysfunction.**

A, B   LysoTracker-stained acidic compartments in zebrafish larvae brains at 3 and 5 dpf. Total signal was increased across the brains of *ups16* crispants at 5 dpf, and abnormal, large puncta observed in the optic tectum from 3 dpf (arrows).
C   LysoTracker signal colocalizes with the astroglial reporter *slc1a2b:citrine* and does not appear to overlap with the pan-neuronal reporter *elavl3:GCaMP5G*.
D   The puncta indicated in A exist within cells expressing the *mpeg1:EGFP* transgene, revealing microglia with a grossly altered cellular morphology.
E   Microglia at 3 dpf contain an increased number of acidic compartments as compared with controls.
F   Global GFP-Lc3 signal is increased in unstimulated *ups16* embryos (left panel), with clusters of puncta present particularly along the midline of the optic tectum (right panel, magnified view) and anterior hindbrain.
G   Magnified view of the anterior hindbrain showing numerous GFP-Lc3 puncta in the *ups16* embryo that do not colocalize with LysoTracker.

Data information: Scale bars 100 μm in (A, F left), 30 μm in (C), 20 μm in (F right), and 10 μm in (D, G). Bar graphs represent data as mean ±SEM. Statistical comparisons by Mann–Whitney *U*-tests. **$P < 0.01$; ***$P < 0.001$. $n = 6$ each for 3dpf, 3 (ctrl) and 12 (vps16) for 5 dpf in (B). $n = 28$ (ctrl) and 14 (vps16) in (E). All confocal images representative of a minimum of 6 embryos each.

Meanwhile, Lc3-positive puncta were almost entirely absent from controls. The Lc3-positive puncta in crispants did not appear to colocalize with LysoTracker (Fig 8G). While this observation is compatible with a fusion defect, quenching of GFP in autolysosomes (Fig 6H) may also contribute. As the CMV promoter drives ubiquitous expression that is not tied to demand for Lc3, the global increase in GFP signal seen in *vps16* embryos may also reflect an impaired turnover of Lc3 protein. Thus, we cannot rule out the possibility that Lc3-positive puncta seen in the *vps16* embryos may represent protein aggregates rather than true autophagosomes. Potential fusion defects were further characterized by time-lapse imaging of CMV:*gfp-lc3* embryos in combination with LysoTracker (Movies EV1 and EV2). No fusion events between Lc3- and Lyso-Tracker-positive puncta were noted for the duration within the locations observed. However, these videos revealed that *vps16* microglia—identified by location, morphology, and LysoTracker phenotype—retain vigorous phagocytic capacity and that defects arise from downstream processing of the phagosome.

## Discussion

Here, we report a new disease caused by a bi-allelic intronic variant in *VPS16*, characterized by severe developmental regression, delayed myelination, brain atrophy with thin corpus callosum, coarse facial features, hypertrichosis, dysostosis multiplex, neutropenia, and feeding difficulties. Support for the pathogenicity of the variant comes from multiple lines of evidence: (i) It is exclusively found in homozygosity in affected patients of two independent families while absent in control populations (Fig 2A and B); (ii) we experimentally show that it impairs the normal splicing of *VPS16* (Fig EV2); (iii) patient-derived fibroblasts have reduced levels of VPS16 protein and a cellular phenotype that is complemented by exogenous *VPS16* expression (Fig 4); (iv) the clinical phenotypes of our patients resemble those of patients with mutations in *VPS33A*, to which VPS16 directly binds; and (v) specific disruption of the zebrafish homolog of *VPS16* recapitulates key patient phenotypes *in vivo* and reveals multiple underlying cellular phenotypes including lysosome accumulation and reduced myelination. Nevertheless, both of our patients were born to consanguineous parents and additional recessive variants may contribute to their phenotypes. For example, patient B additionally suffers from biotinidase deficiency, although this was diagnosed by screening and supplemented. The same patient is also homozygous for a variant of unknown significance in *SCN2A*, that is absent in the gnomAD database. Pathogenic mutations in *SCN2A*, which encodes a voltage-gated sodium channel, are mainly linked to early-onset epilepsy syndromes but have also been linked to episodic ataxia, intellectual disability, and autism (Wolff *et al*, 2017; Reynolds *et al*, 2020). While a contribution of the *SCN2A* variant to the neurodevelopmental phenotype of patient B cannot be fully excluded, we find a major impact unlikely for the following reasons: (i) developmental delay in *SCN2A*-related diseases is associated with epileptic activity (Sanders *et al*, 2018; Wolff *et al*, 2019) and our patient never had apparent seizures or pathological EEG; (ii) *SCN2A*-related diseases are normally autosomal dominant and caused by *de novo* pathogenic variants, except for benign familial neonatal or infantile seizures (OMIM #607745, Wolff *et al*, 2017). Both parents of our patient are heterozygous for the *SCN2A* variant but have neither epilepsy nor intellectual disability; and (iii) there are extensive phenotypic similarities with patient A, who does not carry this variant.

Variants in *VPS16* have also been linked to primary dystonia (Cai *et al*, 2016; Steel *et al*, 2020), for example, a homozygous c.156C>A missense variant segregated with isolated, adolescent-onset primary dystonia in a large consanguineous Chinese family (Cai *et al*, 2016). A possible explanation for the discrepancies in patient phenotypes is that the c.156C>A variant and resulting VPS16$^{N52K}$ protein act via a different mechanism that does not affect the levels of HOPS/CORVET complexes. Indeed, we found that the VPS16$^{N52K}$ protein could rescue protein levels of VPS33A (Fig EV3C and D) and the transferrin uptake defect (Fig 4C and D) in our patient-derived fibroblasts similar to wild-type VPS16. Some fish species also naturally carry a lysine in the same position (Fig EV3E), arguing against a conserved importance of this residue in the function of HOPS/CORVET complexes. Heterozygous loss-of-function mutations in VPS16 are also overrepresented in a cohort of patients with primary dystonia, with onset between 3 and 50 years of age (Steel *et al*, 2020). However, both the homozygous c.156C>A variant and heterozygous loss-of-function variants are observed in presumably healthy individuals in the GnomAD repository (Karczewski *et al*, 2020) and none of the parents of our patients report dystonia. Heterozygous *VPS16* loss-of-function mutations thus appear to have reduced penetrance and can possibly be viewed as a genetic risk factor for dystonia. This suggests that *VPS16* can be added to the growing list of genes in which bi-allelic variants lead to an LSD, while heterozygous carriers have increased risk for a movement disorder (Ysselstein *et al*, 2018): the most well-known example being *GBA1*, which is responsible for Gaucher's disease but also confers risk for Parkinson's disease in heterozygotes (Aflaki *et al*, 2017).

There is significant overlap in the clinical phenotypes of our patients and the previously described patients with a homozygous R498W missense mutation in *VPS33A* (Dursun *et al*, 2017; Kondo *et al*, 2017; Pavlova *et al*, 2019). Both syndromes present with developmental regression, skeletal abnormalities (dysostosis multiplex), joint contractures, hematological impairments, organ enlargement, and MPS-like dysmorphism (see Table 1 for a summary comparison). Together with the fact that VPS33A directly binds to VPS16 (Baker *et al*, 2013; Graham *et al*, 2013; Wartosch *et al*, 2015), this suggests a common disease mechanism for both conditions. The R498W mutation does not affect the binding of VPS33A to VPS16 (Kondo *et al*, 2017). Instead, the mutation has been shown to destabilize the protein (Pavlova *et al*, 2019), leading to reduced cellular levels of VPS33A as well as HOPS subunits VPS18 and VPS41 (VPS16 was not analyzed). Our finding that a quantitative reduction in HOPS/CORVET complexes below a certain threshold is sufficient to cause disease supports this hypothesis. However, there may be clinical and biochemical differences between the two conditions. For example, VPS33A patients suffer from renal dysfunction and cardiomyopathy, which we have not observed in our patients (although a sibling to one patient reportedly died from cardiac complications of unknown cause). These differences may reflect the limited number of patients reported so far, and as the patients have been young, it cannot be excluded that cardiorenal complications may develop later in life. Alternatively, there may be cell type-specific differences in the relative levels of specific HOPS/CORVET subunits. For example, cells in which VPS16 is rate-limiting for HOPS/CORVET assembly would be more sensitive to its reduction. A similar mechanism may explain why patients with mutations in VPS11 and VPS41 present with clinical phenotypes that are more restricted to the brain (Table 1).

By analyzing patient-derived fibroblasts, we identified several cellular phenotypes that may help to explain how the intronic mutation causes disease. Although the use of fibroblasts as a model system has certain limitations—for example, they show high variability and may not present cell type-specific phenotypes—our ability to complement phenotypes by re-expression of *VPS16* allowed for well-controlled experiments. First, we identify a defect in endosomal compartments by studying the uptake and intracellular trafficking of transferrin (Fig 4). Endocytosed transferrin is normally recycled to the plasma membrane either directly from early endosomes or indirectly via recycling endosomes (Maxfield & McGraw, 2004). Blockade of recycling pathways has been reported to instead direct transferrin to lysosomal compartments (Jonker *et al*, 2018). Our findings showing that increased levels of transferrin end up in acidic compartments (Fig 4G and H), thus supporting a defect in endosomal recycling pathways. A defect in these pathways is further supported by the observed increase in transferrin receptor levels (Fig 4E and F), although further studies are needed to elucidate the underlying mechanism. Whether these defects have consequences for the cellular iron uptake in patients is also not known. The mild microcytic anemia observed in both patients, together with the elevated levels of soluble transferrin receptors measured in the serum of one patient despite continuous oral iron supplementation and an otherwise normal iron status (Fig EV1C), might indicate that. However, other causes such as increased erythropoiesis cannot be ruled out.

The patient-derived fibroblasts also showed an accumulation of autophagosomes (Fig 6), likely caused by decreased HOPS-mediated fusion of lysosomes and autophagosomes and similar to HeLa cells depleted of VPS16 (Wartosch *et al*, 2015). Consistently, a lower proportion of LC3-labeled puncta in patient-derived cells were acidified (Fig 6H). However, the relative increase in lipidated LC3 after addition of bafilomycin A1 was unchanged (Fig 6E). In light of the observations that patient cells maintain residual levels of wild-type VPS16 (Fig 3B and C) and capacity to form autolysosomes (Fig 6I), we hypothesize that the increased number of autophagosomes at least in part may compensate for reduced autophagosome–lysosome fusion rates to maintain a near-normal net autophagic flux (Fig 6E). It is less obvious how defective vesicle fusion may result in the observed accumulation of acidic compartments (Fig 5). A hypothetical model has, however, been put forward in the context of VPS33A patients, which share this phenotype (Kondo *et al*, 2017; Pavlova *et al*, 2019). Lysosomes depend on the continuous fusion with late endosomes—either by transient kiss-and-run or complete fusion events (Luzio *et al*, 2007)—to regenerate endolysosomes that are the principal compartment for acid hydrolase activities (Luzio *et al*, 2007; Huotari & Helenius, 2011; Bright *et al*, 2016). Reduced HOPS complex levels resulting from pathogenic variants in *VPS16* or *VPS33A* may impair regeneration of functional endolysosomes. Over time, this may lead to accumulated defects in lysosomal functions. Future experimental studies will be needed to determine how rate-limiting levels of HOPS/CORVET complexes affect flux between different endolysosomal compartments.

The zebrafish model provides additional insights into the role of *vps16* in development and disease. Zebrafish deficient in *vps16* are reminiscent of other HOPS component mutants (Maldonado *et al*, 2006; Schonthaler *et al*, 2008; Thomas *et al*, 2011), which lack swim bladders and show pigmentation abnormalities in skin melanocytes and retina, thus corroborating the phenotypes of impaired HOPS complex function in zebrafish. Pigmentation abnormalities suggest dysfunction of lysosome-related organelles in melanocytes, as is also seen in *buff* mice that carry a spontaneous D251E mutation in *Vps33a* (Suzuki *et al*, 2003). While this mouse model has been considered a model of the Hermansky–Pudlak syndrome (HPS; OMIM PS203300), no genes found to cause HPS in humans have been linked to HOPS/CORVET complex functions and neither our patients nor the *VPS33A* R498W patients (Dursun *et al*, 2017; Kondo *et al*, 2017; Pavlova *et al*, 2019) display symptoms of HPS such as hypopigmentation and bleeding diathesis. Moreover, further characterization of the *buff* mouse line has shown significant motor learning deficits not common to HPS (Zhen & Li, 2015), while mutation of a human-associated HPS gene in zebrafish (*dtnbp1*) caused pigmentation deficits but minimal effects on CNS acidification (Chen *et al*, 2018). Thus, while VPS and HPS variants both lead to abnormalities in LROs that may contribute to common pigmentation deficits in animals, these are distinctly different conditions.

Importantly, the zebrafish recapitulated key patient observations such as reduced myelination. Given that myelin production only begins at 3 dpf in zebrafish embryos, it is most likely that the phenotype we see in the embryos indicates delayed and/or defective myelinogenesis. This may contribute to leukodystrophy and/or hypomyelination later on, but further studies are required to assess these possibilities. The zebrafish model also confirms the accumulation of both acidic compartments and autophagosomes seen in patient-derived fibroblasts. Moreover, it expands these findings by suggesting that glial cells are primarily affected, in

particular microglia. While the endocytic capacity of microglia appeared intact in *vps16* embryos, the accumulation of acidic compartments in these cells suggests defective lysosomal processing. Microglia are known to be involved in a number of key neurodevelopmental processes, such as the clearance of apoptotic cell bodies and pruning of synapses (Marín-Teva *et al,* 2004; Paolicelli *et al,* 2011), and defects in this lineage have been linked to a wide range of neurological diseases, including LSDs. Recent studies of other leukodystrophies, including lysosomal metachromatic leukodystrophy and Krabbe disease, as well as a recently discovered leukodystrophy involving the absence of microglia, have suggested that microglial abnormalities may contribute to their pathogenesis (Oosterhof *et al,* 2019; Weinstock *et al*, 2020; Wolf *et al*, 2020). The early finding of microglial abnormalities in our zebrafish model supports this hypothesis, although further studies are needed to determine the contribution of this cell type, and others, to the pathology of *vps16* disruption.

In summary, we describe a pathogenic intronic variant in VPS16 as a cause of a new LSD and thereby expand the list of "MPS-Plus" diseases caused by reduced HOPS/CORVET complexes. The zebrafish and fibroblast models are well suited to further increase our understanding of how this defect disrupts endosomal, lysosomal, and autophagosomal functions in different cell types to cause the observed clinical phenotypes.

# Materials and Methods

### Clinical sequencing

For patient A, DNA extraction from blood was performed as described (Darin *et al,* 2018). Whole-genome sequencing (WGS) was performed on a HiSeq X Ten system (Illumina) and reads were trimmed and mapped against the hg19 reference genome using CLC Biomedical Server (Qiagen) essentially as described (Darin *et al,* 2018). Variants were filtered using a workflow developed at the Centre for Clinical Genomics at Sahlgrenska University Hospital against genes in the Lysoplex *in silico* panel that contains 891 genes involved in lysosomal, endocytic, and autophagic pathways [Table EV1C; (Di Fruscio *et al,* 2015)].

Exome sequencing of patient B was performed in a trio setup using SureSelect V6 r2 enrichment kit (Agilent) and a HiSeq4000 sequencer (Illumina) at the Cologne Center for Genomics (CCG, University of Cologne, Germany). The data were analyzed using the Varbank 2 pipeline 3.5 (https://varbank.ccg.uni-koeln.de/varbank2). Reads were mapped to the human genome reference build hg38 using the BWA-mem alignment algorithm (Li, 2013). Duplicates were removed using Picard (http://broadinstitute.github.io/picard). GATK (v3.6, (McKenna *et al,* 2010)) was used for local indel realignment around short insertions or/and deletions (INDEL), base quality score recalibration and to call single nucleotide variants (SNV) and short INDELs. Rare variants (MAF ≤ 0.001 in gnomAD, (Lek *et al,* 2016)) were filtered with GATK for high quality (SNV filter: Allele Frequency 0.25; QualByDepth (QD) ≥ 2; FisherStrand (FS) ≤ 60; ReadPosRankSumTest (ReadPosRankSum) ≥ −8, RMS Mapping Quality (MQ) ≥ 40; Mapping Quality Rank Sum Test (MQRankSum) ≥ −12.5; and INDEL Filter: QD ≥ 2, FS ≤ 200, ReadPosRankSum ≥ -20). Splice variants were filtered by maximum

entropy scores (MaxEntScan, Min wt score > 2, Score Change < −15% (Yeo & Burge, 2004)). The DeNovoGear software was used to identify de novo mutations (Ramu *et al,* 2013) De novo probability 0.2)). Pipeline-related false positives were further reduced by taking advantage of an in-house database, containing 7022 epilepsy exomes. Variants were filtered using the Lysoplex *in silico* panel (Di Fruscio *et al,* 2015). De novo, compound heterozygous, and homozygous variants were considered for further analysis.

### RNA isolation and cDNA analysis

RNA was isolated using RNeasy Plus Mini kit (Qiagen) according to the manufacturer's protocol. The RNA concentrations were measured using the NanoDrop 2000 spectrophotometer (Thermo Fisher Scientific). cDNA was synthesized using the SuperScript VILO cDNA Synthesis Kit (Invitrogen) according to the manufacturer's protocol. Primers *VPS16*-ex22F-5′-CCTGGCAGATTTGGAAGATT-3′ and *VPS16*-UTRR-5′-TAGCAGGGCTGTCTGGTTTT-3′ positioned in exon 22 and 3′-UTR, respectively, of the VPS16 gene (NM_022575.3) were used for PCR amplification of part of VPS16 exon 22 to 3′-UTR (PCR conditions available upon request). PCR products were visualized on agarose gels for verification of specific PCR product, and Sanger sequenced in both directions using PCR primers and BigDye v.3.1 (Life Technology). Fragments were separated on a 3130xL Genetic Analyzer (Applied Biosystems), and the sequences were analyzed in Sequence scanner (Thermo Fisher Scientific). Primers were designed using the Primer3 software (Primer3 v. 0.4.0). For quantitative RT–PCR, cDNA was generated using the High-Capacity cDNA Reverse Transcription Kit (Thermo Fisher Scientific) and analyzed using the PrimeTime Gene Expression Master Mix (IDT) using a 7900HT Real-Time PCR System (Applied Biosystems). Primers and FAM-labeled, ZEN/IBFQ-quenched hydrolysis probes for *VPS16* exon 7–9 (primer 1, primer 2, probe: 5'-TCTCCTTCACCTACCGACAC, 5'-GCACCAGACCATCTGCTTT, 5'-TT GAGTGATGCTGTCCCCATCCAG), VPS16 exon 22–24 (5'-TTTTCCAA GAGCAAGAAATCACC, 5'-CATCGCCAACAAGAAGCAAAG, 5'-CAGA TCTCCACAAAAGGCAGGTAGCC), *VPS11* (5'-GATGGCTCCAACTTTC CAATG, 5'-GCCCAGATTTTCATGCAGTATG, 5'-ATCGTGGTTGCCCT TGCTGTAGA), and *VPS33A* (5'-CCGAGTGAACCTAAACGTGT, 5'-GG GTCCAGTTAGGTATTCATCC, 5'-ACTATTGCCTTGCTTCCTGCGC) were obtained from IDT. *POLR2A* (5'-CAGTTCGGAGTCCTGAGTC, 5'-TCGTCTCTGGGTATTTGATGC, 5'-ACTGAAGCGAATGTCTGTGA CGGAG) was used to normalize results by the ΔΔCt method.

### Molecular cloning

A cDNA clone encoding the full-length human VPS16 (NM_022575.4) was obtained from the DNASU Plasmid Repository at Arizona State University. The open reading frame was amplified by PCR and cloned downstream of the ubiquitin promoter in a lentiviral expression plasmid by isothermal assembly (HiFi DNA Assembly kit, NEB). The c.156C>A variant (encoding the N52K mutant) was introduced by PCR and assembled similarly.

### Lentiviral production

Lentivirus was produced in HEK293/T17 cells (ATCC, CRL-11268) and were grown in DMEM-GlutaMAX (Thermo Fisher Scientific,

cat#31966021) supplemented with 10% fetal bovine serum (Sigma-Aldrich). Cells were co-transfected with 3rd generation lentiviral packaging plasmids and either control (empty) or *VPS16*-expressing vectors using calcium phosphate. Media was replaced 1 h prior to transfection to media supplemented with 25 mM chloroquine (Sigma-Aldrich) and again replaced 4–6 h later to medium supplemented with 10 mM HEPES (Thermo Fisher Scientific, cat#15630080). Conditioned media containing lentiviral particles were harvested 24–36 h later, cleared by centrifugation at 1,500 $g$ for 10 min, and snap-frozen in aliquots until use.

### Patient fibroblast culture

Dermal fibroblasts from patients and age-matched subjects clinically investigated for non-LSD conditions and used as controls were cultured in DMEM-GlutaMAX high glucose, pyruvate (Thermo Fisher Scientific, cat#31966021) supplemented with 10% fetal bovine serum (FBS; HyClone SV30160.03), at 37°C in a humidified atmosphere with 5% CO2. Cells were passaged using TrypLE Select (Thermo Fisher Scientific, cat#12563011). Fibroblasts were transduced with lentiviruses by adding 1/10 volume of conditioned media from virus-producing HEK293 cells (see above).

For analysis of autophagic flux, cells grown to 100% confluence were washed twice in 0.1 M PBS and incubated with 200 nM Bafilomycin A1 (Cayman Chemicals) or vehicle (dimethylsulfoxide, Sigma-Aldrich D2438) for 2 h in serum-free culture medium before harvest. For proteasome inhibition, MG132 was added to a final concentration of 1 μM to cells 0, 30, or 120 min before harvest.

### Acidic compartment labeling

Fibroblasts were grown on coverslips until 70–80% confluence, washed twice with PBS, and serum-starved for 2 h. Acidic compartments were labeled with 50 nM LysoTracker Deep Red (Invitrogen, L12492) for 30 min in serum-free DMEM, followed by 30-min chase in serum-free medium. Samples were then washed, fixed, and stained as described below.

Labeling of acidic compartments in zebrafish embryos was performed as previously described (Kuil *et al*, 2019). Briefly, zebrafish larvae were incubated in the dark at 28°C for 40 min in open 1.5-ml tubes with 100 μl LysoTracker™ Red DND-99 (Thermo Fisher, Waltham, MA) diluted to a final concentration of 10 μM in E3 containing 200 μM PTU. The media was then replaced with fresh E3-PTU to wash away excess dye, and the fish were mounted for imaging after at least 20 min.

### Quantitative determination of Transferrin uptake

Confluent fibroblasts in 6-well plates were washed with PBS, serum-starved for 2 h followed by feeding with 25 μg/ml Transferrin-488 (Invitrogen) in DMEM for 30 min at 37°C. The cells were washed briefly with PBS followed by a brief wash with Acidic Wash Buffer (50 mM Glycin 150 mM NaCl pH 3) followed by another wash with Acidic Wash Buffer for 3 min at RT to remove surface-bound Transferrin. After a final wash with PBS, cells were lysed RIPA buffer (110 μl/well) with protease inhibitors (see below) for 20 min on ice under repeated vortexing. Cells were

scraped off and sonicated for 5 min in an ice-bath. Lysates were freeze-thawed and subsequently cleared by centrifugation at 20,000 $g$ for 20 min at 4°C. Fluorescence in the supernatant was measured with a fluorometric Microplate Reader Flouroscan FL (Thermo Fisher) excitation filter 485 and emission filter 538. The amount of uptake was normalized to the total protein level (Pierce BCA Protein Assay Kit, Thermo Scientific).

### LC3 puncta analysis

Fibroblasts were transfected with the ptfLC3 plasmid (Addgene #21074), using an Amaxa 4D nucleofector (Lonza) set at program CM138 and solution P3 (20 μl per 250,000 cells and 1 μg of plasmid). Transfected cells were seeded onto glass-bottom cell culture dishes (Greiner Bio-one; cat# 627870) and imaged 1–2 days later using a A1plus confocal system (Nikon Instruments) equipped with a chamber to keep cells at 37°C in a humidified 5% $CO_2$ atmosphere (OKOlab). Images were acquired using a 40X/NA1.15 water immersion objective or a 60X/NA1.4 oil immersion objective. Puncta were analyzed on single optical sections using the NIS-Elements AR software package (v. 5.21.01; Nikon Instruments) with the General Analysis add-on. Images were median-filtered to reduce noise and puncta were contrast-enhanced by the "Detect regional maxima" function. Red and green+red puncta were automatically detected based on a set threshold. The fraction of autophago-lysosomes in each cell was calculated as the number of (red-[green+red])/red puncta.

### Cell lysis and immunoblotting

Fibroblasts were washed once with PBS and lysed in RIPA buffer (Tris 50 mM pH 8.0, NaCl 150 mM; sodium dodecyl sulfate 0.1%; sodium deoxycholate 0.5%; Triton X-100 1%) complemented with protease inhibitor cocktail (Roche, 11697498001) and 2 mM phenylmethylsulfonyl fluoride (Sigma-Aldrich). Lysates were cleared by centrifugation at 20,000 $g$ at 4°C, mixed with 5× sample Buffer (0.31 M Tris–HCL pH 6.8; 10% sodium dodecyl sulfate; 0.05% bromophenol blue; 0.5 M glycerol; 10% DTT), resolved by SDS–PAGE using 16% polyacrylamide gels for LC3 and 4–20% gradient gels (Bio-Rad, 5671095) for other proteins. Gels were transferred onto Protran NC nitrocellulose membranes (GE Healthcare). Membranes were blocked for 1 h in 5% milk (Bio-Rad) in TBS with 0.05% Tween 20 (TBS-T) and incubated with primary antibodies in 5% milk/TBS-T overnight at 4°C under orbital shaking. The next day, membranes were washed three times 10 min at RT using TBS-T, incubated for 1 h at RT with secondary antibodies anti-mouse 680RD (Li-Cor, 925-68070) and anti-rabbit 800CW (Li-Cor, 925-32211), washed as before, and visualized using an Odyssey CLx system (Li-Cor). Immunoblotted bands were quantified by densitometry using Image Studio 5.2 software (Li-Cor).

Primary Antibodies used for immunoblot analysis were VPS16 (BioSite, 17776-1-AP; used at 1:1,000 dilution), VPS33A (BioSite, 168961-AP; 1:1,000), VPS11 (BioSite, ASJ-376977; 1:1,000), LC3 (MBL, PM036; 1:1,000), and β-Actin (Sigma-Aldrich, A5441; 1:5,000).

### Transferrin feeding for confocal imaging

Transferrin uptake in fibroblasts was visualized using fluorescently labeled Alexa488-conjugated Transferrin (Invitrogen). Cells grown

to 80% confluency on coverslips in 24-well plates were washed with PBS, serum-starved for 2 h, then incubated with 25 µg/ml Transferrin-488 in DMEM for 30 min at 37°C. The cells were washed with PBS followed by a brief wash with Acidic Wash Buffer (50 mM Glycin 150 mM NaCl pH 3) and incubation 3 min at RT with Acidic Wash Buffer to remove surface-bound Transferrin, washed again one time with PBS, and fixed using 4% PFA/ 4% Sucrose in PBS. To label lysosomal compartments, cells were incubated with LysoTracker together with Transferrin or immunolabeled for LAMP2 following fixation (see below).

## Immunofluorescence

Fibroblasts were grown on coverslips, serum-starved for 2 h—for consistency with transferrin feeding experiments—and fixed with 4% paraformaldehyde, 4% sucrose in 0.1 M PBS for 15 min, washed three times with PBS, and permeabilized with 0.1% (v/v) Triton X-100 PBS for 10 min. After additional washes, unspecific binding sites were blocked with 2% normal goat serum and 1% BSA in PBS for 1 h at RT. Primary antibodies EEA1 (eBioscience, 14-9114-82; used at 1:1,000 dilution); LAMP2 (SouthernBiotech, 9840-01; 1:50) were incubated 1 h in 3% BSA and washed as before. Secondary Alexa Fluor antibodies (Invitrogen) were incubated at a 1:1,000 dilution for 1 h at RT. Nuclei were counterstained with DAPI nuclear stain (Sigma-Aldrich) together with the secondary antibody. Coverslips were mounted with Prolong Gold Anti-Fade mounting medium (Invitrogen).

## Confocal microscopy

Cells were visualized by confocal microscopy using a A1 plus confocal system (Nikon Instruments). Images were acquired using a 40X/NA1.15 water immersion objective or a 60X/NA1.4 oil immersion objective and analyzed using NIS-Elements AR software package (v. 5.21.01; Nikon Instruments). The General Analysis add-on was used to define binary masks for each channel based on thresholds defined by signal intensity and size. Fluorescence intensity and the number of objects were measured for Transferrin, EEA1, LysoTracker, and LAMP2, respectively, in maximum intensity projections in z. The number of puncta was normalized to number of cells, as determined by counting nuclei. At least six fields per sample and three samples per condition were analyzed. Co-localization was assessed by Pearson coefficient for co-localization measured using the coloc2 plugin for ImageJ/Fiji (Schindelin et al, 2012).

Zebrafish larvae were anesthetized using tricaine (0.016%) prior to being mounted in 1.8% low melting point agarose as previously described (Kuil et al, 2019). The imaging dish was covered with HEPES-buffered E3 containing tricaine during imaging. In vivo fluorescence imaging employed a Leica SP5 intravital microscope with a 20X/NA1.0 water-dipping objective (Leica Plan-Apochromat) using 488 and 561 nm laser lines. Videos were collected using a 6-min time interval. Images were analyzed using ImageJ software (Schindelin et al, 2012).

## Zebrafish lines and husbandry

Zebrafish stocks are maintained using standard conditions in accordance with EU Directive 2010/63 Article 33 and Annex III. Adults are kept at 28.5°C under a 14:10 light:dark cycle and used for spawnings 1× per week. This study employed the following zebrafish lines: AB, Tg(mbp:eGFP-caax), Tg(mpeg1:GFP), Tg(CMV:EGFP-map1lc3b), TgBAC(slc1a2b:Citrine), and Tg(elavl3:GCaMP5G) (Almeida et al, 2011; Ellett et al, 2011; Ahrens et al, 2013; He et al, 2014; Kuil et al, 2019). To facilitate imaging myelination without the need for PTU, the Tg(mbp:eGFP-caax) line has been crossed into a nacre background and only full nacre embryos were selected for imaging.

## CRISPR/Cas9 mutagenesis

CRISPR/Cas9 mutagenesis was performed largely as previously described (Kuil et al, 2019). The sgRNAs were generated from annealed primers, one containing a minimal T7 RNA polymerase promoter, the target sequence, and a tail-primer target sequence (5' – aattaatacgactcactataGGGAGTGGCGACACAGACCTgttttagagcta gaaatagc – 3'), and the other comprised of a generic tail-end primer (Vejnar et al, 2016). CRISPRscan was used to identify the target sequence on exon 15 of vps16 (New Haven, CT) (Moreno-Mateos et al, 2015). PCR fragments containing a T7 promoter and gRNA sequence were generated using the FastStart™ High Fidelity PCR System (Sigma, St. Louis, MO) as previously described (Moreno-Mateos et al, 2015). The mMESSAGE mMACHINE™ T7 ULTRA Transcription Kit (Invitrogen, Carlsbad, CA) was used according to the manufacturer's instructions to transcribe and purify gRNAs from PCR fragments. RNA quality and concentration were assessed by agarose gel electrophoresis and NanoDrop™ (Thermo Fisher, Waltham, MA) spectrophotometry. The SP-Cas9 plasmid used for the production of Cas9 protein was obtained from Addgene (plasmid #62731), deposited by Niels Geijsen (D'Astolfo et al, 2015). Cas9 nuclease was synthesized as described (D'Astolfo et al, 2015). For injections, 600 ng gRNA was mixed with 4 ng of Cas9 nuclease and adjusted to a total volume of 6 µl with 300 mM KCl. To facilitate visualization, 0.3 µl of 0.5% phenol red (Sigma-Aldrich) was added just prior to injections. Approximately 1.5 nl of this mix was injected into fertilized zebrafish oocytes. Genomic DNA surrounding the gRNA target site was sequenced using Sanger sequencing (Fwd: 5' – TATGGGTTGTCAGTCAGCTTT – 3', Rev: 5' – TTGGAATGG GTGTGTGTGAATG – 3') and indel efficiency and frequency were determined using the online tool TIDE (Brinkman et al, 2014). Controls were injected with Cas9 loaded with an inactive gRNA (5' – aattaatacgactcactataCACGGCTCAAGCATTCGGTGgttttagagcta gaaatagc – 3'). Mutagenized zebrafish injected with vps16 gRNA are referred to as crispants throughout this report.

## Statistical analyses

Data were analyzed from three experiments using GraphPad (Prism; version 8). Power analysis was not performed. Zebrafish embryos of the same clutch were randomly chosen to receive vps16 gRNA or control injections; subsequent blinding was not possible due to obvious differences in pigmentation and swim bladder development resulting from disruption of vps16. In fibroblast experiments, image acquisition and analysis were performed with the conditions blinded to the experimenter.

Statistical significance was evaluated using unpaired two-tailed Student's t-tests, one-way ANOVA with Holm–Sidak's or Dunnett's

### The paper explained

#### Problem

Lysosomal storage diseases (LSDs) comprise a family of rare, inherited disorders characterized by defects in the lysosome, a compartment within cells that degrades large molecules. We identified two unrelated patients suffering from early developmental delay and abnormalities in the brain, blood, and bone that are suggestive of an LSD. Yet, biochemical and genetic tests for known LSDs failed to identify the underlying cause.

#### Results

Genome sequencing revealed that both patients carried the same mutation in *VPS16*. This gene encodes a common subunit of two protein complexes, called HOPS and CORVET, that are important for the fusion of endosomal and lysosomal vesicles within cells. By analyzing skin cells from the patients, we found that the mutation caused a dramatic reduction in both VPS16 protein and other HOPS/CORVET subunits. Consequentially, these cells showed an accumulation of lysosomes and defects in the intracellular trafficking of endosomes and lysosomes. These defects were corrected when VPS16 levels were restored. Moreover, we generated a genetic zebrafish model that recapitulated the accumulation of lysosomes seen in patient-derived cells and the decreased myelination observed in patients, while also suggesting that glial disturbance plays a substantial role in the disease.

#### Impact

This study describes a new LSD and provides further insights into the mechanisms of diseases that result from defects in HOPS/CORVET complexes.

multiple comparison tests or Mann–Whitney *U*-tests, as described in figure legends. Data presented in bar graphs are shown as mean ± standard error of the mean (SEM). For exact *n*- and *P*-values, see Appendix Table S1.

### Ethics statement

The study was approved by the Regional Ethics Committee at the University of Gothenburg, Sweden, and conducted according to the World Medical Association Declaration of Helsinki and the Department of Health and Human Services Belmont Report. Signed informed consent was obtained from the legal guardians of the patients.

## Data availability

This study includes no data deposited in external repositories.

**Expanded View** for this article is available online.

### Acknowledgments

The authors are grateful to the patients and their families. FHS is supported by grants from the Swedish State Support for Clinical Research (ALFGBG-797811 to FHS), the Swedish Research Council (dnr 2017-03331) and by the Knut and Alice Wallenberg Foundation via the Wallenberg Centre for Molecular and Translational Medicine at the University of Gothenburg, Sweden. TvH is supported by an Erasmus University Rotterdam fellowship. TSB is supported by the Netherlands Organisation for Scientific Research (ZonMW Veni, grant 91617021), a NARSAD Young Investigator Grant from the Brain & Behavior Research Foundation, an Erasmus MC Fellowship 2017, and Erasmus MC Human Disease Model Award 2018. LeS is supported by a LEaDing fellowship from the European Union's Horizon 2020 research and innovation program, under the Marie Skłodowska-Curie grant agreement No 707404. KM and JG are supported by grants from the German Research Foundation (Ga354/14-1).

### Author contributions

Study conception: KM, JAC, TvM, TSB, and FHS; Data collection: KS, KM, CM, LaS, ES, MB, JAC, and FHS; Clinical data analysis: KS, KM, CM, LaS, ES, MB, JAC, FHS, ND, and JG; Zebrafish model and data collection: LeS; Zebrafish data analysis: LeS and TvH; Data collection and analysis from fibroblasts: DK, LMG, and FHS; Reagents: LaS and LA; Manuscript writing: KS, KM, LeS, DK, TvH, JAC, and FHS. Manuscript revision for important intellectual content: All authors. Study supervision: FHS.

### Conflict of interest

The authors declare that they have no conflict of interest.

### For more information

- European Reference Network for Hereditary Metabolic Disorders: https://metab.ern-net.eu
- Online Mendelian Inheritance in Man: https://www.omim.org
- Genome aggregation database (gnomAD): https://gnomad.broadinstitute.org
- GeneMatcher: https://genematcher.org

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
