## [Review Process File · EMBO Molecular Medicine]

Bi-allelic VPS16 variants limit HOPS/CORVET levels and cause a mucopolysaccharidosis-like disease

Kalliopi Sofou, Kolja Meier, Leslie Sanderson, Debora Kaminski, Laia Montoliu Gaya, Emma Samuelsson, Maria Blomqvist, Lotta Agholme, Jutta Gärtner, Chris Mühlhausen, Niklas Darin, Tahsin Stefan Barakat, Lars Schlotawa, Tjakko van Ham, Jorge Asin Cayuela, and Fredrik Sterky
DOI: [10.15252/emmm.202013376](https://doi.org/10.15252/emmm.202013376)

Corresponding author: Fredrik Sterky (fredrik.sterky@gu.se)

Review Timeline:

Submission Date:	1st Sep 20
Editorial Decision:	17th Sep 20
Revision Received:	12th Feb 21
Editorial Decision:	11th Mar 21
Revision Received:	18th Mar 21
Accepted:	19th Mar 21

Editor: Zeljko Durdevic

Transaction Report:

17th Sep 2020

Dear Prof. Sterky,

Thank you for the submission of your manuscript to EMBO Molecular Medicine. We have now received feedback from the three reviewers who agreed to evaluate your manuscript. As you will see from the reports below, the referees acknowledge the interest of the study but also raise some concerns that should be addressed in a major revision. The focus of the revision should be on the detailed assessment of the endocytic trafficking and autophagy in the patient fibroblasts and zebrafish model.

Addressing the reviewers' concerns in full will be necessary for further considering the manuscript in our journal, and acceptance of the manuscript will entail a second round of review. EMBO Molecular Medicine encourages a single round of revision only and therefore, acceptance or rejection of the manuscript will depend on the completeness of your responses included in the next, final version of the manuscript. For this reason, and to save you from any frustrations in the end, I would strongly advise against returning an incomplete revision.

We realize that the current situation is exceptional on the account of the COVID-19/SARS-CoV-2 pandemic. Therefore, please let us know if you need more than three months to revise the manuscript.

I look forward to receiving your revised manuscript.

Yours sincerely,

Zeljko Durdevic

***** Reviewer's comments *****

Referee #1 (Remarks for Author):

The work by Sofou presents a novel and interesting human mutation in the HOPS/CORVET subunit VPS16. The authors provide compelling and conclusive evidence that this biallelic mutation in VPS16 is pathogenic. This conclusion is robustly supported by rescue experiments of mutant patient fibroblasts with wild type VPS16 as well as the generation of a zebrafish model where the Vps16 gene defect recapitulate key aspects of the human disease. I think this is an important contribution with a well-balanced discussion of the data in the context of other mutations in HOPS

complex subunits. However, while the work is solid there are some ideas put forward by the authors that deserve further experimentation as the evidence is not strong to support some of the authors contentions.

My most substantial criticism is the idea that normal morphology of EEA1 positive compartments is sufficient to state that VSP15 mutants have normal endosome fusion. I do not agree with this statement. If the authors intend to assess the function of early endosomes, I think that dynamic assessment of cargo traffic through endosomes should be tested. I suggest the authors study ligand-dependent EGF receptor degradation and transferrin receptor recycling as a way to assess if indeed there is a defect in endosome traffic in patient cells.

In addition I would like to encourage the authors to consider the following

- 1) I missed a discussion and comparison of the patients phenotypes with those found in the buff mouse mutant. In particular, could the authors contrast/discuss if the patients may have features of Hermansky-Pudlak syndrome. This question is prompted by the pigmentation and swim bladder phenotypes in zebrafish.
- 2) Do patients have defects in pigmentation as compared to the parents and do these patients have defective platelet function? The only reported platelet assessment is their number, which is normal.
- 3) Please spell out the title acronym MPS-like.
- 4) The rationale of studying the VPS16N52K is interesting but the studies are limited to complex assembly. If the authors wish to make a case with this mutant, then they should expand their studies to some of the lysotraker and endosome studies proposed above to make a solid case about the pathogenicity of this mutation. Otherwise, they should eliminate these findings from the manuscript and avoid discussion of this mutant.
- 5) Figure EV5E should be part of figure 5 or the whole figure EV5 should be brought into the main body of the manuscript.

Referee #2 (Remarks for Author):

Sofou and colleagues describe two patients with MPS-like phenotype harboring the same homozygote intronic variant affecting splicing of VP16 a member of the HOPS/CORVET complex. Although only two cases are described in this manuscript, the observation is important because it strengthen the concept, put forward in previous studies, that defects in components of the HOPS/CORVET complex cause lysosomal abnormalities and a neurological and dysmorphic phenotype.

Main points

- 1) Regarding the clinical descriptions:
 - a. Are hand X-rays available for patient A? In patient B there appears to be metacarpal pointing that is a feature of MPS.
 - b. Was there any corneal clouding in the patients?
 - c. Was chitotriosidase elevated also in patient B?
 - d. Was patient A analyzed directly by WGS without performing exome sequencing first? Please explain.
 - e. Growth parameters are only shown at birth. Include length, weight, and OFC at the latest available clinical evaluation.

- f. Testing of the unaffected sibling of family 2 for the variant would be important.
- g. A cautionary note should be added in the discussion about the cardiomyopathy: both cases are relatively young and late onset of the cardiomyopathy cannot be excluded.
- h. It would be extremely useful and informative if the authors include in Table 1 also the other previously described patients carrying mutations in the HOPS/CORVET complex.

2) The SCN2A variant p.Asp1999Tyr is not present in gnomAD. This information should be included. Although the patient does not have epilepsy, hypomyelination and thin corpus callosum are reported in SCN2A disease. Therefore, the authors should elaborate more on the reasons for excluding SCN2A as responsible for the phenotype.

3) In the Western blot in fig. 3B, there are unspecific bands very close to the band that should correspond to VPS16. Moreover, the truncated protein is not detected. To confirm this finding the authors should incubate fibroblasts with proteasome inhibitor in the attempt to visualize the truncated protein. It would also be interesting to evaluate whether protein levels of VPS33A and VPS11 are rescued by the proteasome inhibitor.

4) There is no control for the efficacy of lentivirus-mediated transduction of fibroblasts shown in Fig. 4. Where is the data on accumulation of LysoTracker-stained compartment after Bafilomycin A1 treatment?

5) The analysis of the lysosomal phenotype requires further investigation. It is unclear why there is an increase in the number of LysoTracker-positive vesicles with no apparent defects in autophagy or endocytosis. Experiments aimed at characterizing these pathways are superficial and more in-depth analyses should be performed. For instance, to assess whether endosome-lysosome fusion is affected, the authors should follow and quantify lysosomal delivery of endocytosed cargoes (e.g. fluorescently labeled BSA or Tf). Similarly, analysis of autophagosome-lysosome fusion should be more carefully examined by evaluating LC3-LAMP1 co-localization and by using the well-established GFP-RFP-LC3 probe. Finally, lysosomal degradative capacity also needs better evaluation, by assessing degradation of autophagy cargoes (such as p62) and by evaluating cathepsin activity (e.g. magic red assay or WB evaluation of cathepsin cleavage).

6) The apparent absence of autophagy/endocytosis defects in VPS11-defective cells may be caused by the use of patient fibroblasts. Although the choice of this cell line is well-understandable, patient fibroblasts are often characterized by a high variability and do not always display disease-relevant phenotypes. These limitations should be discussed.

7) The zebrafish model appears a valuable tool to study VPS11 function in vivo. However, only a very limited analysis of the lysosomal phenotype was performed in this system. A deeper characterization of lysosomal function or autophagy defects, using some of the approaches described above, is needed to better characterize the cell-type specificity and relevance of these pathways to the disease pathogenesis.

8) Co-staining of LysoTracker-positive vesicles with lysosomal markers (e.g. LAMP1, Cathepsins) is required in order to claim that the "lysosomal" compartment is expanded in VPS11-defective cells and tissues.

Minor issues

1. In the abstract: please state that 'both patients were homozygous for the same intronic variant'.

2. Results: 'myelopoetic' should read 'myelopoietic'.
3. Fig. 1F refers to patient A according to the figure legend but it is cited for patient B in the text.
4. For patient B, fig. 1F (spine MRI) is placed in the wrong place when referring to the bone abnormalities.
5. Include in Supplementary the genes sequenced through the panels in patients A and B.
6. I suggest using 'Control 1' and 'Control 2' instead of 'Control A' and 'Control B' to avoid confusion with 'Patient A' and 'Patient B'. In figure 4 D, the authors used 'control 2' and 'patient 1' that is inconsistent with the labeling used in other experiments.

Referee #3 (Comments on Novelty/Model System for Author):

Patients have one specific mutation which cause abnormal splicing in mRNA. Zebrafish model has nonspecific ins/del mutations. Please see "reviewer's comments".

Referee #3 (Remarks for Author):

Comments:

The manuscript from Sofou et al. describes two patients with intronic variant in VPS16 gene that impairs normal mRNA splicing. Subsequently, the authors report accumulation of acidic compartments in skin fibroblasts of patients and zebrafish model of disease. This research appears to be well described and shown data seems convincing. It is a first-time description of patients with MPS-like phenotype caused by mutation in VPS16 gene.

Major points:

(Cases)

1. Authors mentioned in the supplementary material that urinary glycosaminoglycans was not increased or just slightly increased in both patients. Information for glycosaminoglycans (fractions, amounts) is important for discussing that this new disease is MPS-like. Please provide additional levels of glycosaminoglycans in patient derived samples such as skin fibroblast or plasma to convince the readers that this disease is MPS-like or mucopolysaccharidosis-like, in which urinary glycosaminoglycans don't show obvious increase.
2. From the context of this manuscript, the authors seem to argue for similarities between their new disease and MPS (by Dursun 2017 Clin Dysmorphol, Kondo 2017 Hum Mol Genet, and recently reviewed by Vasilev 2020 Int J Mol Sci). MPS shows severe clinical phenotype leading to early death within 2 years of their age. Please discuss about the differences of "severity" of these diseases including the information of life expectancy if possible.
3. Information of Chinese patients with other homozygous VPS16 mutation (Cai 2016 Sci Rep) should be listed in the table EV1, and properly discussed about the molecular mechanisms that explain different phenotypes raised from same gene with different mutations. In the discussion section, authors speculated that HOPS/CORVET assembly was not affected in the Chinese patients with c.156C>A mutations. In Cai's paper HOPS/CORVET assembly was not tested. HOPS/CORVET assembly is not tested in this paper. In Kondo's paper HOPS/CORVET assembly was normal and patients showed MPS-like phenotype, although it is caused by the mutation in the other molecule VPS33A.

(Genetic identification)

4. Regarding to the identification of causative gene VPS16, authors shortly described that no pathogenic mutation was found by the gene panel of known lysosomal storage diseases, and finally detected this specific mutation in the VPS16 gene by whole genome sequencing, for example. However, authors should clearly describe the processes how they reached to this specific VPS16 mutation in both patients and families. The reason why authors focused on autosomal recessive trait is unclear. Because both families were in the condition of consanguineous marriage, many candidate genes/variants for autosomal recessive diseases will be listed first. How many candidates appeared, and how authors select more possible genes, and how they finally reached to VPS16? Were Patient A and Patient B analyzed independently, and both happened to led to the same mutation in VPS16 gene this time?

5. Authors describe that the specific mutation c.2272-18C>A was not found in the GnomAD database. Do they mean that the variant occurred in the Family A and Family B independently, and never observed in the general population worldwide?

(molecular analyses)

6. The authors state that the disease results from impaired HOPS/CORVET complex assembly/reduced level of complexes, but there is no evidence for this. They just separately checked level of other subunits and they haven't checked "assembly" of these tethering complexes.

7. Authors checked level of other HOPS/CORVET subunits, VPS33A and VPS11. Observing the protein level of VPS18 may be helpful for understanding the effect of HOPS/CORVET formation on the disease, because VPS18 is also a direct interactor of VPS16.

8. Quality of the data from Fig 4D immunoblot is low, and 4E quantification should also be refined. Authors revealed that autophagic flux is not impaired in patient-derived fibroblasts, but from the representative figure shown in 4D, Patient 1 (probably Patient A? and "Control 2" also be "Control B"?) seems to be showing "reduced" autophagic flux. In the research of Jiang et al. 2014, MBoC, siVPS16 treatment showed residual VPS16 level and reduced autophagic flux activity. Please provide clear image of LC3 immunoblotting, because this information is quite important for the manuscript.

(zebrafish model)

9. Authors engineered vps16 KO in zebrafish. It is unclear whether this fish model adequately recapitulate the disease in human, because 15% of residual protein exists by normal splicing in human patients, however ins/del mutations that may disrupt vps16 were identified in the fish model. Additionally, detection of vps16 protein level would be needed to validate zebrafish model.

10. Please discuss whether "myelination is delayed" or "once-established myelin is broken down, i.e. leukodystrophy", and its mechanistical link with LysoTracker accumulation. Was MR spectroscopy performed on this zebrafish (or impossible)?

Minor points:

1. Typo in Introduction section: "...MPSs include coarse facial features, skeletal deformities, short statue, hepatosplenomegaly...". Statue?

2. Throughout the manuscript, representative images and quantified graphs don't accord. Please

use actually representative images, or re-quantify images to revise graphs.

3. Figure 3B,3D,3E, 3F: For easy understanding, it would be better to show relative values normalized with one control as 1 (or as 100%).

4. Figure 4A: It is nonsense to use Green and White for merged data, because Green dots will be hidden behind the White structure. Also please discuss the necessity of merged data (EEA1+Lysotracker).

5. Figure 4D: Name of samples (Control 2, Patient 1) not match to names presented in Figure 4E (Control B, Patient A): also pointed out in the major comment 8.

6. Figure 3C, EV5B: Authors provided schematic illustration of HOPS/CORVET complexes. Does VPS33A or VPS33A-VPS16 subcomplex directly bind with VPS41 or VPS8 in HOPS/CORVET complexes? Please indicate the references of original figure, and illustrate accurately depending on previous reports (Balderhaar 2013 J Cell Sci, Hunter 2018 J Mol Biol, etc).

Authors' response to the reviewers' comments

In the following, we cite the reviewers' comments in full in *italics*, and provide our response in **bold** typeface.

Referee #1:

The work by Sofou presents a novel and interesting human mutation in the HOPS/CORVET subunit VPS16. The authors provide compelling and conclusive evidence that this biallelic mutation in VPS16 is pathogenic. This conclusion is robustly supported by rescue experiments of mutant patient fibroblasts with wild type VPS16 as well as the generation of a zebrafish model where the Vps16 gene defect recapitulate key aspects of the human disease. I think this is an important contribution with a well-balanced discussion of the data in the context of other mutations in HOPS complex subunits. However, while the work is solid there are some ideas put forward by the authors that deserve further experimentation as the evidence is not strong to support some of the authors contentions.

My most substantial criticism is the idea that normal morphology of EEA1 positive compartments is sufficient to state that VSP15 mutants have normal endosome fusion. I do not agree with this statement. If the authors intend to assess the function of early endosomes, I think that dynamic assessment of cargo traffic through endosomes should be tested. I suggest the authors study ligand-dependent EGF receptor degradation and transferrin receptor recycling as a way to assess if indeed there is a defect in endosome traffic in patient cells.

Response: We appreciate this suggestion and have added several experiments to assess transferrin traffic through the endosomal compartment, summarized in a new figure (Fig 4). The results of those experiments identified a defect in the dynamic endocytosis of the patient cells. We have revised our conclusions and the discussion accordingly.

In addition I would like to encourage the authors to consider the following
1) I missed a discussion and comparison of the patients phenotypes with those found in the buff mouse mutant. In particular, could the authors contrast/discuss if the patients may have features of Hermansky-Pudlak syndrome. This question is prompted by the pigmentation and swim bladder phenotypes in zebrafish.

Response: Hermansky-Pudlak Syndrome (HPS, OMIM [PS203300](https://doi.org/10.1016/j.ceb.2004.05.001)) is characterized by reduced pigmentation/albinism, bleeding diathesis, pulmonary fibrosis, granulomatous colitis, and renal failure. The decreased pigmentation and bleeding diathesis have been related to defects in melanosomes and platelet dense granules, respectively, which are both lysosome related organelles (Dell'Angelica et al 2004; <https://doi.org/10.1016/j.ceb.2004.05.001>). The buff mouse mutant, which is

characterized by a melanosome defect and carries a D251E missense mutation in *Vps33A*, has been considered a model for HPS. In humans, however, variants in 10 different genes have been found to cause HPS but none of those are directly linked to HOPS/CORVET complexes. Conversely, neither of our patients – nor any of the previously reported *VPS33A* R498W patients – display symptoms of HPS such as hypopigmentation and bleeding diathesis. To what extent these discrepancies may reflect inter-species differences or divergent effects of specific mutations is unknown.

In zebrafish, swim bladder development depends on the production and distribution of surfactant from lysosome-related organelles (LROs) (<https://doi.org/10.1534/genetics.117.300621>), while pigmentation depends on LRO-related melanosomes. Of note, many proposed animal models of HPS, including the *buff* mouse, have been identified through forward genetics approaches that focused initially – and often primarily – on pigmentation defects. While bleeding time was also assessed and found to be increased in the initial characterization of the *buff* mouse (Chen et al 2018; <https://dx.doi.org/10.1073%2Fpnas.0237292100>), the authors commented that it was a less severe phenotype than seen in other HPS-like mouse models. Since this initial report, further characterization of the *buff* line has shown significant motor learning deficits not common to HPS (Zhen and Li 2015; <https://doi.org/10.1080/15548627.2015.1072669>), while, mutation of a human-associated HPS gene in zebrafish (*dtnbp1*) leads to pigmentation deficits but has minimal effects on CNS acidification (Chen et al 2018; <https://doi.org/10.1534/genetics.117.300621>). Thus, while VPS and HPS variants both lead to abnormalities in LROs that may contribute to common pigmentation deficits in animals, these are distinctly different conditions. Thus, the proposed models of HPS that are not linked to known HPS disease genes in humans may require careful reevaluation.

2) *Do patients have defects in pigmentation as compared to the parents and do these patients have defective platelet function? The only reported platelet assessment is their number, which is normal.*

Response: Neither patient displayed defects in pigmentation (Fig 1A) or bleeding diathesis. Because platelet numbers were normal (patient A) or recovered (patient B) and there were no clinical signs of impaired hemostasis, additional assessment of platelet function was not performed.

3) *Please spell out the title acronym MPS-like.*

Response: we have rephrased the title “Bi-allelic VPS16 variants limit HOPS/CORVET levels and cause a mucopolysaccharidosis-like disease” as it otherwise exceeds the journal’s maximum length of 100 characters.

4) *The rationale of studying the VPS16N52K is interesting but the studies are limited to complex assembly. If the authors wish to make a case with this mutant, then they*

should expand their studies to some of the lysotraker and endosome studies proposed above to make a solid case about the pathogenicity of this mutation. Otherwise, they should eliminate these findings from the manuscript and avoid discussion of this mutant.

Response: We agree and have included the N52K mutant in the functional test of transferrin uptake (Fig 4C-E). We find that the N52K mutant can rescue the defect in transferrin uptake of the patient-derived cells, similar to wildtype VPS16 and have added this to the discussion.

5) Figure EV5E should be part of figure 5 or the whole figure EV5 should be brought into the main body of the manuscript.

Response: We have moved Fig EV5E to the main Fig 7B.

Referee #2:

Sofou and colleagues describe two patients with MPS-like phenotype harboring the same homozygote intronic variant affecting splicing of VP16 a member of the HOPS/CORVET complex. Although only two cases are described in this manuscript, the observation is important because it strengthen the concept, put forward in previous studies, that defects in components of the HOPS/CORVET complex cause lysosomal abnormalities and a neurological and dysmorphic phenotype.

Main points

1) Regarding the clinical descriptions:

a. Are hand X-rays available for patient A? In patient B there appears to be metacarpal pointing that is a feature of MPS.

Response: A hand X-ray of patient A at 10 months of age was normal, which has been added to the supplementary patient description. We regrettably do not have a later X-ray available and do not find it justified to perform for the purpose of this publication without a medical need for the patient.

b. Was there any corneal clouding in the patients?

Response: No. No obvious corneal clouding was observed in either of the patients.

c. Was chitotriosidase elevated also in patient B?

Response: Chitotriosidase was not measured in patient B who is unfortunately no longer available for additional blood sampling.

d. Was patient A analyzed directly by WGS without performing exome sequencing first? Please explain.

Response: No. As was stated in the results section, patient A was first investigated by a gene-panel for known LSDs. We have further clarified this by including a new Table EV1, which lists the included genes. When the results of this investigation turned out negative, the patient was subject to whole-genome sequencing, which is now clinical routine at our center.

e. Growth parameters are only shown at birth. Include length, weight, and OFC at the latest available clinical evaluation.

Response: Growth parameters at the last encounter have been included in the supplementary extended patient descriptions.

f. Testing of the unaffected sibling of family 2 for the variant would be important.

Response: Predictive genetic testing of clinically unaffected children is generally not possible in Germany by law and due to ethical standards. Thus, the unaffected siblings of family 2 were not tested for the variant.

g. A cautionary note should be added in the discussion about the cardiomyopathy: both cases are relatively young and late onset of the cardiomyopathy cannot be excluded.

Response: We agree and have added this point to the discussion.

h. It would be extremely useful and informative if the authors include in Table 1 also the other previously described patients carrying mutations in the HOPS/CORVET complex.

Response: We have merged the previous Table 1 and Table EV1 into a new Table 1.

2) The SCN2A variant p.Asp1999Tyr is not present in gnomAD. This information should be included. Although the patient does not have epilepsy, hypomyelination and thin corpus callosum are reported in SCN2A disease. Therefore, the authors should elaborate more on the reasons for excluding SCN2A as responsible for the phenotype.

Response: While we cannot fully exclude a partial contribution of the SCN2A variant to the neurodevelopmental phenotype of patient B, the following facts argue against a major impact: (1) There are striking phenotypic similarities between patient B and patient A; patient A does not carry the SCN2A variant. (2) Developmental regression in SCN2A-related diseases is associated with manifestation or exacerbation of epilepsy (Wolff 2019, <https://doi.org/10.1111/epi.14935>; Sanders 2018, <https://doi.org/10.1016/j.tins.2018.03.011>). Our patient did neither show seizures nor epileptic activity on EEG up to the latest follow-up with 21 months, whereas developmental regression was first noticed at 6 months of

age. (3) The inheritance mode in *SCN2A* related diseases is autosomal dominant and mutations usually occur *de novo*, except from benign familial neonatal or infantile seizures (OMIM #607745, Wolff et al. 2017, <https://doi.org/10.1093/brain/awx054>). Both parents of patient 2 are heterozygous for the *SCN2A* variant but have neither epilepsy nor intellectual disability. To assume a contribution of the *SCN2A* variant of our patient one must assume that this would then be the first case of a *SCN2A* related disease with an autosomal recessive inheritance.

The abovementioned reasons for excluding a major contribution of the *SCN2A* variant to the neurodevelopmental phenotype of the patient were added to the discussion of the revised manuscript.

3) In the Western blot in fig. 3B, there are unspecific bands very close to the band that should correspond to VPS16. Moreover, the truncated protein is not detected. To confirm this finding the authors should incubate fibroblasts with proteasome inhibitor in the attempt to visualize the truncated protein. It would also be interesting to evaluate whether protein levels of VPS33A and VPS11 are rescued by the proteasome inhibitor.

Response: We agree, and have now tested if the proteasomal inhibitor MG132 lead to accumulation of a truncated VPS16 protein. As is shown in Fig EV3A-B, we do not find any truncated isoform following proteasome inhibition.

4) There is no control for the efficacy of lentivirus-mediated transduction of fibroblasts shown in Fig. 4. Where is the data on accumulation of LysoTracker-stained compartment after Bafilomycin A1 treatment?

Response: We are using the same lentivirus that gives robust expression of VPS16 as assessed by western blot (Fig 3). This admittedly does not demonstrate whether few cells express high levels or all cells express moderate amounts. However, we have used numerous constructs in the same lentiviral vector in both human skin fibroblasts and other cell types, and consistently found the latter, that transduction is near-100% efficient (see example below), although levels of expression within each of these cells may vary between different constructs. This is further supported by the consistent rescue seen in microscopy experiments (e.g. Fig 5A-B). As we could assess the average expression by immunoblotting for VPS16, but the antibody did not perform well in immunocytochemistry, and experimental approaches to assess transduction efficiency by microscopy – e.g. by fusing the cDNA with an affinity tag or use a bi-cistronic vector to express a fluorescent protein – comes with its own problems, we chose to omit this control.

Bafilomycin A1 inhibits the lysosomal V-ATPase and thereby blocks the acidification that is a requirement lysosomal maturation and accumulation of the LysoTracker dye. Hence, treating LysoTracker-stained cells with bafilomycin A1 is not informative as it causes the LysoTracker dye to dissipate.

5) The analysis of the lysosomal phenotype requires further investigation. It is unclear why there is an increase in the number of lysotracker-positive vesicles with no apparent defects in autophagy or endocytosis. Experiments aimed at characterizing these pathways are superficial and more in-depth analyses should be performed. For instance, to assess whether endosome-lysosome fusion is affected, the authors should follow and quantify lysosomal delivery of endocytosed cargoes (e.g. fluorescently labeled BSA or Tf). Similarly, analysis of autophagosome-lysosome fusion should be more carefully examined by evaluating LC3-LAMP1 co-localization and by using the well-established GFP-RFP-LC3 probe. Finally, lysosomal degradative capacity also needs better evaluation, by assessing degradation of autophagy cargoes (such as p62) and by evaluating cathepsin activity (e.g. magic red assay or WB evaluation of cathepsin cleavage).

Response: We have added several of the suggested experiments to further assess the cellular phenotypes:

- **Feeding of fluorescent transferrin to study the trafficking of transferrin through the endosomal compartment (Fig 4C-H)**
- **A more detailed assessment of autophagosomes and autolysosomes using the GFP-RFP-LC3 probe in fibroblasts (Fig 4F-H)**
- **Assessment of autophagosomes and autolysosomes by GFP-LC3 in zebrafish (Fig 8F-G)**
- **Western blot for p62 in fibroblasts (Fig 5E)**
- **Western blot for Cathepsin D in fibroblasts (Fig 6A-B)**

Together, these experiments have substantially expanded the manuscript, lead to new results and partially revised our previous conclusions. In summary, these experiments now clearly demonstrate that the patient-derived cells have a defect in transferrin uptake and intracellular trafficking, as well as an accumulation of autophagosomes which we also find in the zebrafish model. Lysosomal degradative capacity, however, was not significantly impaired. We have revised the manuscript accordingly.

6) The apparent absence of autophagy/endocytosis defects in VPS11-defective cells

may be caused by the to use of patient fibroblasts. Although the choice of this cell line is well-understandable, patient fibroblasts are often characterized by a high variability and do not always display disease-relevant phenotypes. These limitations should be discussed.

Response: We can only agree that fibroblasts are far from being an ideal model system, and now acknowledge this limitation in the discussion. For example, the relatively small effect sizes seen in these cells, in combination with high variability between experiments, hampered our initial assessment of LC3 by immunoblotting. After careful re-assessment and several additional replicates we have now partially revised our results and the conclusion from this experiment. Nevertheless, the intronic genetic variant found in our patients is difficult to model in other cell types, and the ability to rescue phenotypes by simple re-expressing the protein allows us to include an isogenic control. While reprogramming the fibroblasts into induced pluripotent stem cells is possible, it would realistically take at least a year and thus be beyond the scope of the current revision.

7) The zebrafish model appears a valuable tool to study VPS11 function in vivo. However, only a very limited analysis of the lysosomal phenotype was performed in this system. A deeper characterization of lysosomal function or autophagy defects, using some of the approaches described above, is needed to better characterize the cell-type specificity and relevance of these pathways to the disease pathogenesis.

Response: Further analysis of the cell-type specificity of the lysosomal defect in the zebrafish model indicates that it is primarily glial, with no notable overlap with neurons and significant inclusions found within microglia and radial glia. LysoTracker-intense puncta were not identified in myelinating oligodendrocytes of either condition. However, these cells are not typically known for a high lysosomal content and the greater imaging depth required to visualize them may have additionally confounded our ability to detect any differences in this population. To assess differences in autophagy we employed a CMV:GFP-LC3 transgenic line. The ubiquitous expression of GFP-LC3 is not responsive to demand for LC3 but indicates autophagosomes through puncta formation. The *vps16* embryos showed a strong global accumulation of LC3-GFP, possibly indicating defects in protein turnover. Closer examination within the brain showed a high number of LC3-GFP+ puncta in the *vps16* embryos that did not appear to substantially colocalize with LysoTracker signal. Control embryos showed too few LC3-GFP+ puncta to be able to assess differences in proportions of colocalization with LysoTracker.

We have expanded the description of these phenotypes in the new Fig 8 (and Suppl movies 1-2).

8) Co-staining of lysotracker-positive vesicles with lysosomal markers (e.g. LAMP1, Cathepsins) is required in order to claim that the "lysosomal" compartment is

expanded in VPS11-defective cells and tissues.

Response: We have tested if our LysoTracker stained puncta correspond to LAMP2-reactive vesicles, and find a similar degree of co-localization in both patient and control cells (new Fig EV4C-D). We have also quantified the LAMP2-reactive staining in our cells (new Fig EV4C, E).

Minor issues

- 1. In the abstract: please state that 'both patients were homozygous for the same intronic variant'.*
- 2. Results: 'myelopoetic' should read 'myelopoietic'.*
- 3. Fig. 1F refers to patient A according to the figure legend but it is cited for patient B in the text.*

Response: The above points (#1-3) have been corrected in the text.

- 4. For patient B, fig. 1F (spine MRI) is placed in the wrong place when referring to the bone abnormalities.*

Response: The spine MRI shown in Fig 1F is from patient A. We have separated the panels more in the figure to hopefully make this clearer.

- 5. Include in Supplementary the genes sequenced through the panels in patients A and B.*

Response: We have added the genes included in the gene panels in a new Table EV1.

- 6. I suggest using 'Control 1' and 'Control 2' instead of 'Control A' and 'Control B' to avoid confusion with 'Patient A' and 'Patient B'. In figure 4 D, the authors used 'control 2' and 'patient 1' that is inconsistent with the labeling used in other experiments.*

Response: We have corrected the mistake in the revised version of this experiment (new Fig 6).

Referee #3 (Comments on Novelty/Model System for Author):

Patients have one specific mutation which cause abnormal splicing in mRNA. Zebrafish model has nonspecific ins/del mutations. Please see "reviewer's comments".

Referee #3 (Remarks for Author):

Comments:

The manuscript from Sofou et al. describes two patients with intronic variant in

VPS16 gene that impairs normal mRNA splicing. Subsequently, the authors report accumulation of acidic compartments in skin fibroblasts of patients and zebrafish model of disease. This research appears to be well described and shown data seems convincing. It is a first-time description of patients with MPS-like phenotype caused by mutation in VPS16 gene.

Major points:

(Cases)

1. *Authors mentioned in the supplementary material that urinary glycosaminoglycans was not increased or just slightly increased in both patients. Information for glycosaminoglycans (fractions, amounts) is important for discussing that this new disease is MPS-like. Please provide additional levels of glycosaminoglycans in patient derived samples such as skin fibroblast or plasma to convince the readers that this disease is MPS-like or mucopolidosis-like, in which urinary glycosaminoglycans don't show obvious increase.*

Response: Information on glycosaminoglycan amounts and fractions are given in the patient descriptions. Our conclusion that the patients suffer from an MPS-like disease stems from their clinical presentations: coarse facial features, skeletal abnormalities (dysostosis multiplex) and psychomotor retardation, as well as their resemblance with the 'MPS-plus' VPS33 patients (Vasilev et al 2020; <https://doi.org/10.3390/ijms21020421>; Kondo et al 2017; <https://10.1093/hmg/ddw377>; Dursun et al 2017; <https://10.1097/MCD.000000000000149>; Pavlova 2019; <https://10.1093/hmg/ddz077>). Information on the amounts and types of glycosaminoglycans in urine is provided in the supplementary descriptions. In both patients, excretion at early age was normal, but found to be increased in patient A upon repeated investigation. Repeating the analysis also for patient B is not possible as the patient has unfortunately passed away. The lack of initial increase in urine glycosaminoglycans (both patients) but subsequent increase (patient A) is compatible with our hypothesis that partial loss of VPS16 leads to secondary lysosomal dysfunction that over time leads to accumulation of storage material including glycosaminoglycans. Hence, we classify it as 'MPS-like' rather than a new MPS.

2. *From the context of this manuscript, the authors seem to argue for similarities between their new disease and MPS3A (by Dursun 2017 Clin Dysmorphol, Kondo 2017 Hum Mol Genet, and recently reviewed by Vasilev 2020 Int J Mol Sci). MPS3A shows severe clinical phenotype leading to early death within 2 years of their age. Please discuss about the differences of "severity" of these diseases including the information of life expectancy if possible.*

Response: While most VPS33A^{R498W} patients pass away by two years of age, some have lived to four years (Vasilev et al 2020; <https://doi.org/10.3390/ijms21020421>). One of our patients has lived beyond 4

years but the other died at 2 years and 3 months of age. Moreover, one of our families reports having two miscarriages and the had a boy who died at one year of age (Fig 2A-B); as clinical details and molecular testing are lacking we cannot excluded that these incidents are related to the *VPS16* variant. We believe that these two cases are too few to make meaningful comparison to the *VPS33A*^{R498W} patients. Nevertheless, it possible that the *VPS33A*^{R498W} mutation indeed is more “severe” in terms of life expectancy due to the cardiac engagement seen in this disease. As discussed in the manuscript, our patients do not show signs of cardiomyopathy.

3. Information of Chinese patients with other homozygous VPS16 mutation (Cai 2016 Sci Rep) should be listed in the table EV1, and properly discussed about the molecular mechanisms that explain different phenotypes raised from same gene with different mutations. In the discussion section, authors speculated that HOPS/CORVET assembly was not affected in the Chinese patients with c.156C>A mutations. In Cai's paper HOPS/CORVET assembly was not tested. HOPS/CORVET assembly is not tested in this paper. In Kondo's paper HOPS/CORVET assembly was normal and patients showed MPS-like phenotype, although it is caused by the mutation in the other molecule VPS33A.

Response: We have now included the *VPS16*^{N52K} patients in Table 1, although the clinical data in Cai et al (2016; <https://doi.org/10.1038/srep25834>) is limited. More importantly, we have now tested if the *VPS16*^{N52K} protein can functionally rescue the transferrin uptake phenotype of our patient cells (new Fig 4C-D). These results show that the *VPS16*^{N52K} mutant protein can rescue the phenotype equal to the wildtype protein.

(Genetic identification)

4. Regarding to the identification of causative gene VPS16, authors shortly described that no pathogenic mutation was found by the gene panel of known lysosomal storage diseases, and finally detected this specific mutation in the VPS16 gene by whole genome sequencing, for example. However, authors should clearly describe the processes how they reached to this specific VPS16 mutation in both patients and families. The reason why authors focused on autosomal recessive trait is unclear. Because both families were in the condition of consanguineous marriage, many candidate genes/variants for autosomal recessive diseases will be listed first. How many candidates appeared, and how authors select more possible genes, and how they finally reached to VPS16? Were Patient A and Patient B analyzed independently, and both happened to led to the same mutation in VPS16 gene this time?

Response: Both patients were analyzed and diagnosed independently, and subsequently matched via GeneMatcher. A sentence to clarify this has been added. The process of the NGS data analysis is described in detail in the methods part of the paper. De Novo, compound heterozygous and homozygous variants were included in the analysis. Given that the patients

presented with several features of lysosomal storage diseases, analysis of both patients – independently – focused on candidate genes for lysosomal storage disorders by filtering for genes within the Lysoplex gene set. In the revised manuscript we have extended the description of how variants were filtered.

5. Authors describe that the specific mutation c.2272-18C>A was not found in the GnomAD database. Do they mean that the variant occurred in the Family A and Family B independently, and never observed in the general population worldwide?

Response: This is discussed in the manuscript: “There is no established ancestral link between the families, although this seems likely since both report some roots in Turkey”. Absence of the variant within the ~110.000 alleles represented in the GnomAD dataset suggests that the variant is rare but it may be present in some Middle eastern populations that are underrepresented in GnomAD. A sentence to clarify this has been added.

(molecular analyses)

6. The authors state that the disease results from impaired HOPS/CORVET complex assembly/reduced level of complexes, but there is no evidence for this. They just separately checked level of other subunits and they haven't checked "assembly" of these tethering complexes.

Response: As the expression of VPS11 and VPS33A is maintained (Fig 3H) but the proteins levels reduced (Fig 3E-G) we find protein destabilization to be the most plausible explanation. Immunoprecipitations of entire HOPS complexes has been demonstrated in yeast cells but is not realistic from patient material. As our results in all suggest that there are residual levels of HOPS/CORVET complexes also in patient cells, their isolation would likely not be informative. Nevertheless, we acknowledge that we have not directly assessed complex assembly and have thus revised the phrasing to avoid this term and instead refer to reduced levels.

7. Authors checked level of other HOPS/CORVET subunits, VPS33A and VPS11. Observing the protein level of VPS18 may be helpful for understanding the effect of HOPS/CORVET formation on the disease, because VPS18 is also a direct interactor of VPS16.

Response: We have included VPS18 in our analysis (Fig 3G)

8. Quality of the data from Fig 4D immunoblot is low, and 4E quantification should also be refined. Authors revealed that autophagic flux is not impaired in patient-derived fibroblasts, but from the representative figure shown in 4D, Patient 1 (probably Patient A? and "Control 2" also be "Control B"?) seems to be showing "reduced" autophagic flux. In the research of Jiang et al. 2014, MBoC, siVPS16 treatment showed residual VPS16 level and reduced autophagic flux activity. Please provide clear image of LC3 immunoblotting, because this information is quite

important for the manuscript.

Response: We have worked extensively to repeat and refine our immunoblotting for LC3 in fibroblast lysates (which is not trivial, presumably because fibroblasts contain less LC3 than e.g. HeLa cells). By increasing the number of experiments, excluding low quality blots and normalizing the data we were indeed able to detect a rather small yet significant increase in LC3-II in the patient-derived cells (new Fig 6C-D). However, autophagic flux defined as the ratio of +/- addition of bafilomycin A1 was similar (new Fig 6E). We speculate that decreased autophagosome-lysosome fusion causes an accumulation of autophagosomes, and that their increased numbers may contribute to normal flux.

On the discrepancy with siVPS16 cells – the discrepancy is in fact very similar to the results of Kondo et al (2017; <https://doi.org/10.1093/hmg/ddw377>), which side-by-side compare VPS33A patient fibroblasts and siVPS33A HeLa cells. They find an obvious phenotype in the HeLa cells but no phenotype in their patient fibroblasts (see Fig 4B and S3B of Kondo et al). We believe this may reflect differences between HeLa cells and human fibroblasts, but perhaps more importantly that siRNA cause a more dramatic perturbation than the partial loss-of-functions caused by the variants found in patients. In addition, we have used the RFP-GFP-LC3 probe and immunoblotting for p62 to further assess levels of autophagy.

(zebrafish model)

9. Authors engineered vps16 KO in zebrafish. It is unclear whether this fish model adequately recapitulate the disease in human, because 15% of residual protein exists by normal splicing in human patients, however ins/del mutations that may disrupt vps16 were identified in the fish model. Additionally, detection of vps16 protein level would be needed to validate zebrafish model.

Response: The zebrafish model we developed employed a highly efficient Cas9/gRNA protocol that induces very high rates of disruption in the *vps16* gene. The use of CRISPR-mutagenized embryos (crisprants) such as this has been shown to phenocopy full knockouts but itself technically is not a complete KO (Burger et al 2016; <https://doi.org/10.1242/dev.134809>). This allows rapid analysis of effects in F0 clutches and is especially useful in instances where gene disruption may prevent models from reaching sexual maturity (as is the case with embryos that do not develop functional swim bladders). A range of indels is produced upon injection and mutagenesis rates for the Cas9/gRNA complex used in this study are typically above 95%. However, clear phenotypes (including pigmentation, swim bladder, and lysotracker staining abnormalities) are also seen in instances of less efficient mutagenesis.

With regards to detecting vps16 protein in the zebrafish model, antibodies that are cross-compatible with zebrafish are rare, possibly due to different posttranslational modifications, and we are not currently aware of an antibody that can detect zebrafish Vps16. We are confident that the observations we report are specific to HOPS/CORVET disruption (brought on by mutation of *vps16*) and provide valuable insight into the human condition. Additional support is found through similarities seen with other HOPS/CORVET loss-of-function zebrafish models (*vps18*: Maldonado et al 2006; <https://doi.org/10.1111/j.1600-0749.2006.00320.x>; *vps11*: Zhang et al 2016; <https://doi.org/10.1371/journal.pgen.1005848>; *vps39*: Schonthaler et al 2008; <https://doi.org/10.1242/dev.006098>, *vp41*: Sanderson et al. *Brain*, (In Press; doi: 10.1093/brain/awaa459).

10. Please discuss whether "myelination is delayed" or "once-established myelin is broken down, i.e. leukodystrophy", and its mechanistical link with LysoTracker accumulation. Was MR spectroscopy performed on this zebrafish (or impossible)?
Response: As our zebrafish system is embryonic, myelin is only beginning to be laid down at the timepoints we were able to assess. Thus, we can argue that myelinogenesis is defective but are unable to comment at this time as to whether this process is delayed, shows specific pathology, or is actively being degraded. Degradation already at this early state seems unlikely. MRI of zebrafish embryos is not currently informative for this purpose (it's not high-enough resolution).

Regarding a mechanistic link – substantial LysoTracker signal was not detected at the depth of myelinating oligodendrocytes in our zebrafish model so no differences in oligodendrocytes themselves could be assessed. However, oligodendrocytes interact with both astrocytes and microglia during the deposition and maintenance of myelin. Astrocytes have been linked to myelin lipid supply, with astrocyte-specific inhibition of cholesterol metabolism leading to delayed myelination in mice (Camargo et al 2017; <https://doi.org/10.1371/journal.pbio.1002605>). Meanwhile, microglia interact with and refine/prune myelin sheets during development (Hughes and Appel 2020; <https://doi.org/10.1038/s41593-020-0654-2>). Both cholesterol/lipid processing and effective phagocytosis rely on lysosomal machinery and the significant lysosomal defects seen in the microglia and astrocytes of our *vps16* crispants may render these functions lacking or absent, contributing to reduced myelinogenesis.

Minor points:

1. *Typo in Introduction section: "...MPSs include coarse facial features, skeletal*

deformities, short statue, hepatosplenomegaly...". Stature?

Response: The typo has been corrected.

2. Throughout the manuscript, representative images and quantified graphs don't accord. Please use actually representative images, or re-quantify images to revise graphs.

Response: Here, we unfortunately do not understand where this reviewer is referring to. The same reviewer also acknowledges above that the "shown data seems convincing". We have reviewed our data and do not find obvious discrepancies between the images shown and the summary quantifications. Nevertheless, microscopy images have been revised.

3. Figure 3B,3D,3E, 3F: For easy understanding, it would be better to show relative values normalized with one control as 1 (or as 100%).

Response: We have revised the axes.

4. Figure 4A: It is nonsense to use Green and White for merged data, because Green dots will be hidden behind the White structure. Also please discuss the necessity of merged data (EEA1+Lysotracker).

Response: We have revised this and separated the data into two separate figures (new Fig 4A, Fig 5A).

5. Figure 4D: Name of samples (Control 2, Patient 1) not match to names presented in Figure 4E (Control B, Patient A): also pointed out in the major comment 8.

Response: We have corrected this mistake.

6. Figure 3C, EV5B: Authors provided schematic illustration of HOPS/CORVET complexes. Does VPS33A or VPS33A-VPS16 subcomplex directly bind with VPS41 or VPS8 in HOPS/CORVET complexes? Please indicate the references of original figure, and illustrate accurately depending on previous reports (Balderhaar 2013 J Cell Sci, Hunter 2018 J Mol Biol, etc).

Response: The model is derived from Bröcker et al. (PNAS 2012; <https://doi.org/10.1073/pnas.1117797109>). In this model, the VPS33A-VPS16 subcomplex interacts with both VPS18 and VPS41. The schematic in Fig EV5B (now Fig 7A) has been revised for consistency with those in Fig 3.

11th Mar 2021

Dear Prof. Sterky,

Thank you for the submission of your revised manuscript to EMBO Molecular Medicine. I am pleased to inform you that we will be able to accept your manuscript pending the following final amendments:

- 1) We are currently encountering high number of submissions, so that our data editors were not able to process all received manuscripts. Therefore, we will send you the document with data editor's suggestions as soon as our data editors process your manuscript. Please do not submit your revised manuscript before we send you the file with data editor's suggestions. Thank you for your understanding.
- 2) Please address all the referees' points. The additional experiments suggested by the referee #2 are welcomed but not required. If you decide not to perform additional experiments, please address the referee #2 points by discussing the altered autophagic flux in patient cells.
- 3) In the main manuscript file, please do the following:
 - Make sure that all special characters display well.
 - Move M&M section after Discussion.
 - In M&M, include that, in addition to the WMA Declaration of Helsinki, the experiments conformed to the principles set out in the Department of Health and Human Services Belmont Report.
 - Add Table 1 to the manuscript file and remove colour.
 - Add contribution for Jutta Gärtner and Niklas Darin. Also, please specify contribution LS for Lars Schlotawa and Leslie Sanderson (e.g LaS and LeS).
 - Indicate in legends exact n= and exact p= values, not a range, along with the statistical test used. To keep the figures "clear" some authors found providing an Appendix table Sx with all exact p-values preferable. You are welcome to do this if you want to.
 - In M&M, statistical paragraph should reflect all information that you have filled in the Authors Checklist, especially regarding randomization, blinding, replication etc.
- 4) Appendix: As the appendix will not be typeset, please upload a final version without track changes.
- 5) Movies: Please zipp movie legend as a .doc file with each movie file.
- 6) The Paper Explained: Please add it to the main manuscript file.
- 7) Funding: Please make sure that information about all sources of funding are complete in both our submission system and in the manuscript.
- 8) As part of the EMBO Publications transparent editorial process initiative (see our Editorial at <http://embomolmed.embopress.org/content/2/9/329>), EMBO Molecular Medicine will publish online a Review Process File (RPF) to accompany accepted manuscripts. This file will be published in conjunction with your paper and will include the anonymous referee reports, your point-by-point response and all pertinent correspondence relating to the manuscript. Let us know whether you agree with the publication of the RPF and as here, if you want to remove or not any figures from it prior to publication. Please note that the Authors checklist will be published at the end of the RPF.
- 9) Please provide a point-by-point letter INCLUDING my comments as well as the reviewer's reports and your detailed responses (as Word file).

I look forward to reading a new revised version of your manuscript as soon as possible.

Yours sincerely,

Zeljko Durdevic

***** Reviewer's comments *****

Referee #1 (Comments on Novelty/Model System for Author):

State of the art approaches and multipronged genetic validations make this paper exceptional in quality

Referee #1 (Remarks for Author):

The authors have done an outstanding job address my comments. The paper is solid and a great contribution. I would like to suggest a minor interpretation text change. The sentence "Thus, our results indicate that the reduction in HOPS/CORVET complexes caused by the mutation in VPS16 leads to reduced transferrin uptake and a relative accumulation of transferrin/receptor complexes in acidic late endosomal/lysosomal compartments". I suggest the following "Thus, our results indicate that the reduction in HOPS/CORVET complexes caused by the mutation in VPS16 leads to reduced transferrin uptake and a relative accumulation of transferrin/receptor complexes in acidic late endosomal/lysosomal compartments or a recycling-late endosome hybrid organelle."

Referee #2 (Remarks for Author):

The authors have addressed in a satisfactory manner all of my criticisms. My only concern is related to the analysis of the autophagic flux, which does not look conclusive. Patient-derived fibroblasts show an increase in the lysosomal compartment as monitored by lysotracker (Fig 5A,B). This phenotype is also found in brains from Vps16 KO zebrafish (Fig 8 D,E), which show increased LC3-positive puncta (Fig 8 F,G). These data suggest that a defect in autophagosome-lysosome fusion may occur as a consequence of Vps16 loss of function. However, the data shown in Fig 6 seem to exclude this possibility. When I look at the data I am not convinced that autophagy flux is normal in

these cells. Therefore, I strongly suggest to analyze also fibroblasts from patient B and/or to use different VPS16 KO cellular models (such as HeLa or HEK293 cells). For example, the levels of the autophagy substrate p62 seem to be increased in basal condition in cells from patient A and do not decrease upon starvation to the same extent as in control cells (Fig 6A). This behavior would be in line with a defect in autophagy flux. To better analyze autophagy flux, I also suggest to monitor LC3 levels through IF in the same conditions shown in panel C. Finally, the data obtained using the GFP-RFP-LC3 construct (panel F) should be better quantified by monitoring the number of autophagolysosomes (red puncta) relative to total puncta, instead of monitoring the non-acidified autophagosomes.

Referee #3 (Remarks for Author):

The authors performed a lot of work to improve their paper. Extended functional studies for proteasomal inhibition, autophagic flux activity, cathepsin D, Rab11, etc significantly enhanced understanding of paper results. Now they properly checked the endocytic function and revealed defects in the uptake and endosomal trafficking of transferrin and accumulation of autophagosomes (tfLC3 in patient cells and zebrafish). Unfortunately, authors could not increase information about the accumulations of glycosaminoglycans in their patients due to the limitations of patient-oriented materials. However, considering altogether their data, there is enough rationale to conclude, and I believe that the authors revised the manuscript accordingly.

Authors' response to the reviewers' comments

In the following, we cite the reviewers' comments in full in *italics*, and provide our response in **bold** typeface.

***** Reviewer's comments *****

Referee #1 (Comments on Novelty/Model System for Author):

State of the art approaches and multipronged genetic validations make this paper exceptional in quality

Referee #1 (Remarks for Author):

The authors have done an outstanding job address my comments. The paper is solid and a great contribution. I would like to suggest a minor interpretation text change.

The sentence

"Thus, our results indicate that the reduction in HOPS/CORVET complexes caused by the mutation in VPS16 leads to reduced transferrin uptake and a relative accumulation of transferrin/receptor complexes in acidic late endosomal/lysosomal compartments". I suggest the following

"Thus, our results indicate that the reduction in HOPS/CORVET complexes caused by the mutation in VPS16 leads to reduced transferrin uptake and a relative accumulation of transferrin/receptor complexes in acidic late endosomal/lysosomal compartments or a recycling-late endosome hybrid organelle."

Response: We agree and have changed the sentence as suggested.

Referee #2 (Remarks for Author):

The authors have addressed in a satisfactory manner all of my criticisms. My only concern is related to the analysis of the autophagic flux, which does not look conclusive. Patient-derived fibroblasts show an increase in the lysosomal compartment as monitored by lysotracker (Fig 5A,B). This phenotype is also found in brains from Vps16 KO zebrafish (Fig 8 D,E), which show increased LC3-positive puncta (Fig 8 F,G). These data suggest that a defect in autophagosome-lysosome fusion may occur as a consequence of Vps16 loss of function. However, the data shown in Fig 6 seem to exclude this possibility. When I look at the data I am not convinced that autophagy flux is normal in these cells. Therefore, I strongly suggest to analyze also fibroblasts from patient B and/or to use different VPS16 KO cellular models (such as HeLa or HEK293 cells). For example, the levels of the autophagy substrate p62 seem to be increased in basal condition in cells from patient A and do not decrease upon starvation to the same extent as in control cells (Fig 6A). This behavior would be in line with a defect in autophagy flux. To better analyze autophagy flux, I also suggest to monitor LC3 levels through IF in the same conditions shown in panel C. Finally, the data obtained using the GFP-RFP-LC3 construct (panel F) should be better quantified by monitoring the number of autophago-lysosomes (red puncta) relative to total puncta, instead of monitoring the

non-acidified autophagosomes.

Response: We agree that autophagosome-lysosome fusion rates are likely affected in our patient cells and zebrafish model. This would provide an explanation for the observed accumulation of LC3 puncta seen in both patient cells and zebrafish brains, as well as the increased levels of LC3-II seen by WB, and is consistent with the previously demonstrated role for VPS16 in autophagosome-lysosome fusion (Wartosch et al. 2015; <https://doi.org/10.1111/tra.12283>). However, these results are seemingly at odds with the finding that LC3-II increase to a similar extent in patient and control cells upon addition of bafilomycin A1 as this suggests that autophagic flux is normal. We believe these seemingly inconsistent observations can be reconciled by the following arguments: First, it is important to keep in mind that our patient-derived fibroblasts are not knockouts but have residual levels of wildtype VPS16 (Figure 3C). Possibly because of this, they retain the capacity for autophagosome-lysosome fusion (illustrated in Figure 6I). If the availability of HOPS complexes is rate-limiting for autophagosome-lysosome fusion, decreased levels of HOPS complexes will lead to an accumulation of autophagosomes. If autophagosome-lysosome fusion were to follow the law of mass action, such substrate buildup (accumulation of autophagosomes and lysosomes) could at least in part compensate for reduced fusion rates to maintain the net autophagic flux. This idea was put forward in the discussion (“a possible explanation for these findings is that the increased number of autophagosomes can compensate for reduced fusion rates to maintain net autophagic flux”) but was regrettably overlooked in the results section. We have revised the text to clarify this.

To further support this, we have quantified more cells in the GFP-RFP-LC3 experiment (Fig 6G, updated) and also quantified the fraction of acidified autolysosomes in the same experiments, as suggested (new Fig 6H). The fraction of acidified autolysosomes was indeed lower in patient cells, consistent with decreased fusion.

We have also analyzed the number of LC3 puncta in fibroblasts by immunofluorescence. While this showed a similar trend, with increased levels of LC3-reactive puncta in patient cells, we found the signal-to-noise ratio of the immunolabeling unsatisfactory; we could not convince ourselves to what extent the labelled puncta represented actual autophagosomes (the comparably low levels of LC3 expressed by our fibroblasts presumably contributes to this), and therefore decided to not include this data.

VPS16 knockout cells would presumably show a stronger phenotype, but as the focus of our manuscript is to describe disease mechanisms in patients with reduced VPS16 levels, we do not believe that such experiments would add much to our manuscript. Moreover, the effects of VPS16 knockdown on autophagosome-lysosome fusion in HeLa cells have already been reported (Wartosch et al. 2015; <https://doi.org/10.1111/tra.12283>).

Finally, we have also noted that the patient-derived cells appear to show an abnormal response to starvation (Fig 6A-B and 6F-G). Impaired starvation response and mTORC1 signaling has been reported in a preprint studying VPS41 knockout and patient-derived cells carrying VPS41 missense variants (van der Welle et al, bioRxiv doi.org/10.1101/2019.12.18.867333). To what extent the similar mechanisms may result from loss of VPS16 remains to be addressed in future work.

Referee #3 (Remarks for Author):

The authors performed a lot of work to improve their paper. Extended functional studies for proteasomal inhibition, autophagic flux activity, cathepsin D, Rab11, etc significantly enhanced understanding of paper results. Now they properly checked the endocytic function and revealed defects in the uptake and endosomal trafficking of transferrin and accumulation of autophagosomes (tfLC3 in patient cells and zebrafish). Unfortunately, authors could not increase information about the accumulations of glycosaminoglycans in their patients due to the limitations of patient-oriented materials. However, considering altogether their data, there is enough rationale to conclude, and I believe that the authors revised the manuscript accordingly.

We are pleased to inform you that your manuscript is accepted for publication and is now being sent to our publisher to be included in the next available issue of EMBO Molecular Medicine.

Corresponding Author Name: Fredrik H Sterky
Journal Submitted to: EMBO Molecular Medicine
Manuscript Number: EMM-2020-13376